# Retention time prediction using neural networks increases identifications in crosslinking mass spectrometry

Sven H. Giese [1,2,3,5], Ludwig R. Sinn [1,5], Fritz Wegner [1] & Juri Rappsilber [1,4]✉

Crosslinking mass spectrometry has developed into a robust technique that is increasingly used to investigate the interactomes of organelles and cells. However, the incomplete and noisy information in the mass spectra of crosslinked peptides limits the numbers of protein–protein interactions that can be confidently identified. Here, we leverage chromatographic retention time information to aid the identification of crosslinked peptides from mass spectra. Our Siamese machine learning model xiRT achieves highly accurate retention time predictions of crosslinked peptides in a multi-dimensional separation of crosslinked *E. coli* lysate. Importantly, supplementing the search engine score with retention time features leads to a substantial increase in protein–protein interactions without affecting confidence. This approach is not limited to cell lysates and multi-dimensional separation but also improves considerably the analysis of crosslinked multiprotein complexes with a single chromatographic dimension. Retention times are a powerful complement to mass spectrometric information to increase the sensitivity of crosslinking mass spectrometry analyses.

[1] Bioanalytics, Institute of Biotechnology, Technische Universität Berlin, Berlin, Germany. [2] Data Analytics and Computational Statistics, Hasso Plattner Institute for Digital Engineering, Potsdam, Germany. [3] Digital Engineering Faculty, University of Potsdam, Potsdam, Germany. [4] Wellcome Centre for Cell Biology, School of Biological Sciences, University of Edinburgh, Edinburgh, UK. [5]These authors contributed equally: Sven H. Giese, Ludwig R. Sinn. ✉email: juri.rappsilber@tu-berlin.de

Crosslinking mass spectrometry (crosslinking MS) reveals the topology of proteins, protein complexes, and protein–protein interactions[1]. Fueled by experimental and computational improvements, the field is moving towards the analyses of interactomes of organelles and cells[1–3]. The identification of crosslinked peptides poses three major challenges. First, the low abundance of crosslinked peptides compared to linear peptides decreases their chance for mass spectrometric observation. Second, the unequal fragmentation of the two peptides leads to a biased total crosslinked peptide spectrum match (CSM) score[4,5]. Third, the combinatorial complexity from searching all the possible peptide pairs in a sample increases the chance for random matches. These challenges increase from the analysis of individual proteins to organelles and cells.

To address the challenge of low abundance, Crosslinking MS studies routinely rely on chromatographic methods to enrich and fractionate crosslinked peptides[1,2,6]. Essentially all analyses contain at least one chromatographic step, by directly coupling reversed-phase (RP) chromatography separation to the mass spectrometer (LC–MS). Additional separation is frequently employed when more complex systems are being analyzed. Strong cation exchange chromatography (SCX)[7,8] was used for the analysis of HeLa cell lysate[9] or murine mitochondria[10]. Size-exclusion chromatography (SEC)[11] was used to fractionate crosslinked HeLa cell lysate[12] and *Drosophila melanogaster* embryos extracts[13]. Multi-dimensional peptide pre-fractionation was used for the analysis of crosslinked human mitochondria (SCX-SEC)[14] and *M. pneumoniae* (SCX-hSAX)[15]. Such multi-dimensional chromatography workflows can yield in the order of 10,000 CSM at a 1–5% false discovery rate (FDR)[14–17].

The identification of cross-linked peptides from spectra is however still challenged by the uneven fragmentation of the two peptides and the large search space that increase the odds of random matches. This is especially the case for heteromeric crosslinks as the size of their search space exceeds that of self-links, i.e., links falling within a protein or homomer[16]. Typically, database search tools use the precursor mass and fragmentation spectrum for the identification of peptides to compute a single final score for each CSM. For linear peptides, post-search methods such as Percolator[18] have been developed that train a machine learning predictor to discriminate correct from incorrect peptide identification. Percolator uses additional spectral information (features) such as charge, length, and other enzymatic descriptors of the peptide[19] to compute a final support vector machine (SVM) score. Similarly, the crosslink search engine Kojak[20] supports the use of PeptideProphet[21,22] and XlinkX[23] supports Percolator[18], while pLink2[24] and ProteinProspector[4] have a built-in SVM classifier to re-rank CSMs. Although RT data are readily available, none of these tools use the, often multi-dimensional, RT information for improved identification in crosslinking studies. A prerequisite for this would be that retention times could be predicted reliably.

For linear peptides, RT prediction has been implemented under various chromatographic conditions[25–31]. In contrast, RTs of crosslinked peptides have not been predicted yet. A suitable machine learning approach for this could be deep learning[32]. Deep neural networks have been successfully applied in proteomics, for example for de novo sequencing[33] or for the prediction of retention times[29,34] and fragment ion intensities[35]. Deep learning allows encoding peptide sequences very elegantly through, for example, recurrent neural network (RNN) layers. These layers are especially suited for sequential data and are common in natural language processing[32]. RNNs use the order of amino acids in a peptide to generate predictions without additional feature engineering. However, it is unclear how to encode the two peptides of a crosslink.

Moreover, it is also unclear whether the knowledge of RTs could improve the identification of cross-linked peptides. A common scenario for an identified crosslink is that one of its peptides was matched with high sequence coverage, while the other was matched with poorer sequence coverage[4]. Such CSMs, unfortunately, resemble matches where one peptide is correct and the other is false (i.e., a target-decoy match or a true target and false target match). Another consequence of coverage gaps is the misidentification of noncovalently associated peptides as crosslinks[36]. The severity of this coverage issue depends on the applied acquisition strategy[37], crosslinker chemistry[38], and the details of the implemented scoring in the search engine. Nevertheless, assuming RT predominantly depends on both peptides of a crosslink, it could complement mass spectrometric information and thus improve existing scoring routines and lead to more crosslinks at the same confidence (i.e., constant FDR).

In this study, we prove that analytical separation behavior carries valuable information about both crosslinked peptides and can improve the identification of crosslinks. For this we build a multi-dimensional RT predictor for crosslinked peptides based on a proteome-wide crosslinking experiment comprising 144 acquisitions on an Orbitrap mass spectrometer from extensively fractionated peptides of the soluble high-molecular-weight proteome of *E. coli*. We then investigate the benefits of incorporating the derived RT predictions into the identification process. In addition, we demonstrate the value of RT prediction for a purified multiprotein complex using the reversed-phase chromatography dimension only.

## Results and discussion
This section covers (1) a description of the experimental workflow and the motivation, (2) the evaluation of the developed retention time predictor, (3) an interpretability analysis of the deep neural network, (4) an analysis of the RT features and their importance for rescoring, (5) the evaluation of the rescoring results from an *E. coli* lysate, and (6) the evaluation of the rescoring results from a routine crosslinking MS experiment, i.e., the analysis of a multiprotein complex (FA-complex).

**A substantial fraction of crosslinks below the confidence threshold are correct.** Crosslinked peptides belonging to the high-molecular-weight *E. coli* proteome were deep-fractionated along three chromatographic dimensions (hSAX, SCX, and RP). This 3D fractionation approach led to 144 LC–MS runs as some of the 90 fractions contained enough material for repeated analysis. The resulting data were searched with an entrapment database approach (Fig. 1a) leading to 11,196 CSMs (11072 TT, 87 TD, 37 DD, Supplementary Fig. 3) at 1% CSM-FDR, separating self and heteromeric CSMs[16,39,40]. The human entrapment database allows to assess error, independently of the target-decoy approach. This will play a critical role here as *E. coli* decoys will be used for the machine learning-based rescoring (but not for the RT prediction). Judged by a set of peptide characteristic metrics (e.g., peptide length, pI, GRAVY) the human entrapment database resembles the properties of the *E. coli* target database (Supplementary Fig. 4).

Before attempting RT prediction and subsequent complementation of search scores, we investigated the extent of false negatives, approximated here by PPIs present in STRING[41] or APID[42] database. At 1% CSM-FDR, 110 such "validated" (val) protein–protein interactions were identified. 10%, 30%, and 50% CSM-FDR returned 226, 278, and 418 validated PPIs, respectively (Fig. 1b). When raising the CSM-FDR from 1% to 50% we thus saw a nearly 4-fold increase in the detectable number of validated PPIs. In contrast, using a pessimistic approach of semi-randomly

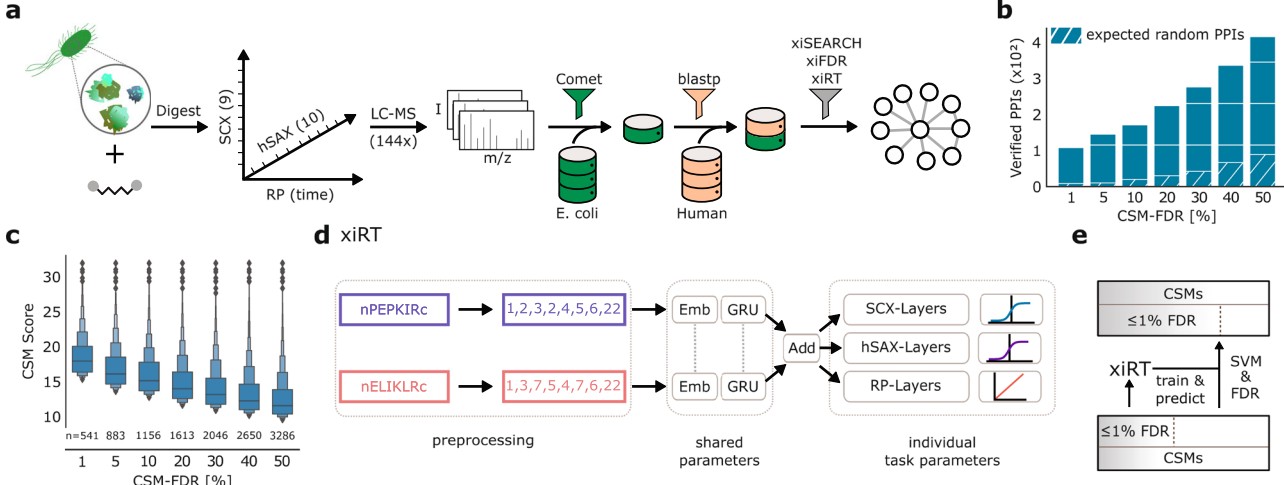

**Fig. 1 Workflow overview. a** Experimental and data analysis workflow. The soluble high-molecular-weight proteome of *E. coli* lysate was crosslinked and the digest sequentially fractionated by strong-cation exchange chromatography (SCX) (9 fractions collected), hydrophilic strong-anion exchange chromatography (hSAX) (10 pools collected), and finally by reversed-phase chromatography (RP) coupled to the MS. The protein database for the crosslink search was created by a linear peptide search with Comet and a sequence-based filter using BLAST. For each *E. coli* protein in the final database (green) a human protein was added as a control (pale orange). **b** Potential for false-negative PPI identifications. Verified PPIs are estimated from matches to the STRING/APID databases. PPIs are computed based on CSM-level FDR. Estimated random hits correspond to the average number of semi-randomly drawn pairs (first protein was randomly selected from the STRING/APID database and the second protein was drawn from the FASTA file). Gained PPIs accentuate the additional information that is available in the data at higher FDR. **c** Decrease of heteromeric CSM scores based on spectral evidence with increasing CSM-FDR. Boxplot shows the median and 50% of the data in the central boxes while each successive level outward represents half of the remaining data. The sample size for each FDR category is given below the boxes. **d** xiRT network architecture to predict multi-dimensional retention times. A crosslinked peptide is represented as two individual inputs to xiRT. xiRT uses a Siamese network architecture that shares the weights of the embedding and recurrent layers. Individual layers for the prediction tasks are added with custom activation functions (sigmoid/linear functions for fractionation/regression tasks, respectively). **e** Rescoring workflow. The predictions from xiRT are combined with xiSCORE's output to rescore CSMs using a linear support vector machine (SVM), consequently leading to more matches at constant confidence. Source data are provided as a Source Data file.

drawing pairs of *E. coli* proteins from the STRING/APID (first protein) and the search database (second protein) yielded purely by chance 10, 22, 44, and 91 overlapping PPIs with STRING or APID for 1%, 10%, 30%, and 50% CSM-FDR cutoffs, respectively. While this shows that loosening the FDR threshold increases validated PPIs also by chance, the actual observed number is much higher (418 versus 91 at 50% CSM-FDR). This means that there is a substantial number of valid PPIs with insufficient match confidence.

The underlying scoring challenge is essential to the identification of peptides in general. The plethora of search engines for linear[43] and crosslinked peptides[44] use spectral characteristics differently for their scoring. In xiSEARCH, the final score is a composite that incorporates spectral metrics such as explained intensity and matched number of fragments. Empirically, we observe a fast decrease in the search engine score (Fig. 1c) with increasing FDR. This indicates that at higher FDRs spectral matching metrics might be suboptimal. Poor spectral quality, inefficient peptide fragmentation, or random fragment matching all influence the search engine score negatively. RT information could complement MS information but this would require accurate RT prediction of cross-linked peptides.

**Accurate multi-dimensional retention time prediction for crosslinked peptides**. RT prediction for crosslinked peptides has not yet been achieved. One reason for this is the challenge of encoding a crosslinked pair of peptides for machine learning. We overcame this here using a Siamese neural network as part of a new machine learning application, xiRT (Fig. 1d), which allowed the incorporation of RTs into a rescoring workflow (Fig. 1e). The Siamese part of the network (embedding layer and recurrent layer) shares the same weights for both peptides. Practically, the

sharing of weights leads to consistent predictions, independent of the peptide order. After the recurrent layer, the two outputs are combined and passed to three subnetworks consisting of dense layers with individual prediction layers (details on the architecture are available in Supplementary Fig. 1). In this multi-task learning setup, the network simultaneously learns to predict the hSAX, SCX and RP RT through a single training step. Multi-task learning can improve the overall performance of predictors by forcing the network to learn a robust representation of the input data[45].

The training and evaluation of xiRT followed a cross-validation (CV) strategy that avoided the simultaneous learning and prediction on overlapping parts of the data (see "Methods" section, Fig. 2a). We used a 3-fold CV strategy where two folds were used for training (excluding 10% for the validation throughout the training epochs) and one fold for testing/prediction. All CSMs with an FDR < 1% were used during the CV. For the remaining CSMs, the best predictor (with the lowest total loss) was used to predict the RTs.

To achieve the best possible prediction performance, hyper-parameters of the network were optimized. Since extensive hyper-parameter optimization on a small data set can lead to overfitting, we initially optimized a large part of hyper-parameters using 20,802 unique linear peptide identifications at 1% FDR. The final parameters for the Siamese network architecture for crosslinks were obtained by a small grid-search (6453 unique peptide-pairs at 1% CSM-FDR; Supplementary Fig. 5).

Using these parameters, we evaluated the learning behavior during the training time (epochs) across the CV folds. The training behavior on the three CV folds was similar and reached a stable trajectory after approximately 15 epochs (Fig. 2b). Based on very similar error trends on validation and training sets, we

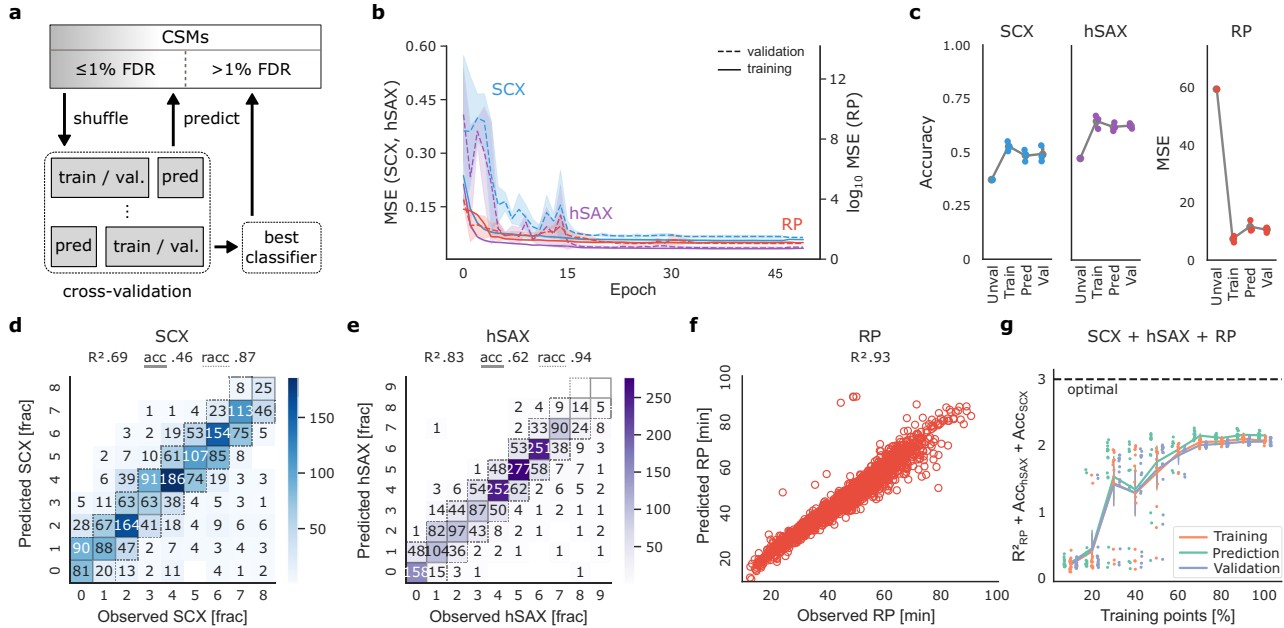

**Fig. 2 Cross-validation of retention time prediction. a** Applied cross-validation (CV) strategy in xiRT. To predict the retention times of CSMs excluded from training, the best CV classifier is used. **b** xiRT performance over training epochs for strong-cation exchange chromatography (SCX, blue), hydrophilic strong-anion exchange chromatography (hSAX, purple), and reversed-phase chromatography (RP, red) prediction with $k = 3$ CV-folds. Shaded areas show the estimated 95% confidence interval with the dashed/solid line representing the mean for the validation/training data, respectively. **c** xiRT performance across different metrics (error bars show standard deviation with the mean as center) for $k = 3$ CV folds. Prediction for the "unvalidated" data was only performed once. **d–f** Prediction results from a representative CV iteration for SCX, hSAX, and RP at 1% CSM-FDR. The achieved $R^2$, accuracy (acc) and relaxed accuracy (racc) are given at the top. **g** Learning curve with increasing number of CSMs, e.g., 10% (645 total CSMs, 387 for training, 43 for validation, 215 for prediction), 50% (3226, 1935, 216, 1075), 100% (6453, 3871, 431, 2151); bars indicate standard deviation with the line representing the mean for the training (red), prediction (green), and validation (blue) data. Source data are provided as a Source Data file.

concluded to have reached a state where neither overfitting nor underfitting occurred. The overall performance across the prediction folds was comparable in terms of accuracy (hSAX: 61% ± 1.1, SCX: 47% ± 1.7) and MSE (RP: 11.58 ± 2.0) (Fig. 2c). Comparing single-task and multi-task configurations of xiRT revealed no significant differences in the prediction accuracy but greatly reduced run times (Supplementary Figs. 6 and 7). Note that we estimated the theoretical boundaries given the ambiguous elution behavior (i.e., peptide elution across multiple chromatographic fractions) for SCX at 65% accuracy and for hSAX at 73% accuracy (Supplementary Table 4 and Supplementary Fig. 8). Most of the predictions showed only a small error, and thus a high relaxed accuracy: for hSAX 94% ± 0.0 and for SCX 87% ± 1.15 of the predictions were within a range of ± 1 fraction (Fig. 2d, e). The overall $R^2_{RP}$ of 0.94 ± 0.01 also showed a predictable relationship for the RP dimension (Fig. 2f). The consistent accuracy and $R^2$ results across CV folds demonstrate reproducible training and prediction behavior which reduces unwanted biases from the different CV folds. In conclusion, RTs of crosslinked peptides can robustly be learned within a data set, making them available as features in a CSM rescoring framework.

It was difficult to compare our RT predictions to other studies which used SCX[46] or hSAX[29] for multiple reasons: (1) there is currently no other model that predicts the RT of crosslinked peptides, (2) the recent SSRCalc[46] study (SCX) for linear peptides used a much larger data set of 34,454 unique peptides and the fractionation was much more fine-grained (30–50 fractions). Similarly, the hSAX[29] study on linear peptides used a much finer fractionation (30 fractions) and a different methodology to encode the loss function during the machine learning. (3) Applied gradients and liquid chromatography conditions can change the elution behavior quite drastically. In our study, the number of

observations was neither for hSAX nor for SCX equally distributed but varied between ~200 and ~2000 CSMs per fraction (Supplementary Fig. 3). Since we employed a partially exponential gradient during the chromatographic fractionation, the degree of peptide separation varied for earlier and later fractions.

Given that we had less data to train on than recent RT predictions of linear peptides, we evaluated how the numbers of observations influenced the prediction accuracy ($R^2_{RP} + Acc_{hsax} + Acc_{scx}$, Fig. 2g). The learning curve showed two important characteristics: first, the prediction performance over CV folds was very reproducible. This means that predictions were robust even with very moderate data quantity. Second, the maximal performance was achieved with ~70–100% of the data points (100% corresponding to 6453 total CSMs, 3871 for training, 431 for validation, 2151 for prediction). Given that a first plateau was reached with 30% of the data, it is unclear if the final prediction accuracy constitutes another local optimum or the limit of the prediction accuracy. The individual task metrics showed that the RP behavior seemed to be easier for the model to learn than the ordinal regression tasks (SCX, hSAX, Supplementary Fig. 9). The RP behavior could be accurately predicted from ~60% of the data points, while the maximum accuracy for hSAX and SCX dimensions was only achieved by using 80–100% of the data. In other words, while using even fewer CSMs might be possible when predicting RP RTs, one would expect a reduced accuracy in the hSAX/SCX dimensions.

An approach to reduce the number of required CSMs would be to leverage the abundantly available data on linear peptides for transfer learning. Indeed, a recent study showed that transfer learning across different peptide identification results works well for linear peptides[34]. We also implemented the option to

pre-train on linear data in xiRT. However, a robust and accurate RT prediction could be achieved on a multiprotein complex crosslinking study (FA-complex, see below) when first training on the *E. coli* CSMs (Supplementary Fig. 10). Another possibility to increase the training data size and robustness during CV is to increase the number of folds, e.g., 5- or 10-fold, at the cost of runtime. Increasing the expedience of xiRT, we also implemented transfer learning for cases when the number of fractions differs between the initial model and the new prediction task.

**Explainable deep learning reveals amino acid contributions.** Using the SHAP package, we set out to explain predictions made by xiRT. For instance, when a specific crosslinked peptide was analyzed, residue-specific contributions towards the predicted RT could be computed (Supplementary Fig. 11). The residues D, E, Y, and F displayed high SHAP values indicating a stronger retention during hSAX separation in a randomly chosen peptide. Looking at a specific crosslinked peptide in SCX (Supplementary Fig. 12), the SHAP values highlighted that K and R were the most important residues contributing towards later peptide elution. As one might expect, crosslinked K residues contributed much less towards later elution times than the stronger charged, unmodified K residues. Investigating the SHAP values for a collection of CSMs revealed additional contributions from W for hSAX and H for SCX while returning hydrophobic residues Y, F, W, I, L, V, and M for RP (Supplementary Fig. 13), revealing residue contributions in crosslinked peptides as seen in the respective analyses of linear peptides[29,46,47]. In summary, the SHAP values were good estimates for the individual RT contributions of the amino acid residues.

Next, we investigated the network architecture and the learned feature representations more closely (Supplementary Note 4). As first analysis, the dimensionality reduced embedding space across the network was analyzed (Supplementary Fig. 14). This revealed that the shared sequence-specific layer already captured the RP properties quite well, while the hSAX and SCX properties were not as clearly captured. As expected, the separation of CSMs according to RT increased the further the features propagated through the network. In the last layer, the RP and hSAX subnetworks reached a very good separation, while in the SCX subtask CSMs remained moderately separated in two dimensions.

**RT characteristics for unsupervised separation of true and false CSMs.** Now that we established the RT prediction of crosslinked peptides, we computed a set of chromatographic features to explore their ability to separate true from false CSMs (Supplementary Table 3). Dimensionality reduction was computed for RP only (13 chromatographic features) and for SCX-hSAX-RP (43 chromatographic features) predictions (Fig. 3a, b). Both chromatographic feature sets revealed good separation possibilities for confident TT (99% true, given 1% CSM-FDR) and TD (100% false) identifications in two-dimensional space. For the RP analysis, the TD *E. coli* CSMs and TT Mix/TD Mix CSMs were enriched in one area of the plot (the lower right part, Fig. 3a). In contrast, the subset of confident TT *E. coli* CSMs were distributed outside this area. As one would expect for two sets of random matches, the CSMs from the entrapment database (TT Mix, TD Mix) closely followed the distribution of TD *E. coli* CSMs. The areas populated by the known false matches were also populated by an equal number of presumably false TT matches. When the features of all three RT dimensions were considered, the separation of true and false CSMs further improved (Fig. 3b). Again, the distributions of TD *E. coli* CSMs and entrapment CSMs behaved similarly. Interestingly, few CSMs that passed the 1% FDR threshold were located in regions dominated by false

identifications. This might identify them as part of the expectable fraction of 1% false-positive identifications. Importantly, the described separation was achieved unsupervised on RT features alone, i.e., without a search engine score or target-decoy labels.

To test the transferability of our findings, we also ran xiRT with unfiltered pLink2 results (Supplementary Note 4 and Supplementary Fig. 15). The prediction performance from *Q*-value-filtered CSMs was similar to the results with xiSEARCH (Supplementary Fig. 15a–c). A two-sided *t*-test between hSAX, SCX, and RP errors for TT and TDs revealed significant differences in the respective error distributions using pLink2 identifications for the RT predictions (Supplementary Fig. 15d). Importantly, the separation of true and false matches in two-dimensional space was also possible with pLink2 identifications (Supplementary Fig. 15e). In summary, xiRT can learn retention times irrespective of the used search engine and the learned chromatographic features alone carry substantial information to separate true from false matches.

To investigate the relevance of multi-dimensional RT predictions for the identification of cross-linked peptides, we first supplemented each CSM with RT features. Then, we performed a semi-supervised rescoring and evaluated the trained SVM model using the SHAP framework. We chose to analyze SHAP values for the 15 most important retention times features for TT observations (FDR > 1%) that were predicted to be a correct TT identification (Fig. 3c). This analysis revealed a similar magnitude for all 15 SHAP values implying that a single feature alone is insufficient to recognize false matches. Notably, the top 5 features contained features from RP, hSAX, and SCX predictions which indicates that each chromatographic dimension carried relevant information for the rescoring. Because 11 of the 15 features were predictions considering only one of the two peptides and not directly derived from peptide-pairs, the predicted RTs displayed a larger error. This analysis suggests that an RT prediction model for linear peptides can add valuable information for crosslink analyses. In general, the model learned mostly that low errors in the RT dimensions indicate true positive identifications. Thus, the model implicitly learned that the RT of a crosslinked peptide should differ from the RT of the individual peptides. This might become useful especially for distinguishing consecutive[48] from crosslinked peptides or when dealing with gas-phase associated peptides[36].

**Rescoring crosslinked peptides enhances their identification.** Before computing a combined score, we compared the CSM scores based on mass spectrometric information (xiSCORE) and RT features (SVM score, Fig. 4a). Both scores largely agreed. Heteromeric CSMs passing 1% CSM-FDR yielded high SVM scores. Also, most target-decoy CSMs achieved a low SVM score (Fig. 4a, right) and a low xiSCORE (Fig. 4a, top). The SVM score distribution of the TDs matched closely the distribution of TTs in the low scoring area, which indicated that they still modeled random TT matches and that overfitting was avoided. Interestingly, the TTs were overrepresented in the low scoring area for the xiSCORE but not for the SVM score, suggesting that true TTs remained hidden among the random matches when using xiSCORE alone. The broad SVM score distribution of TTs indicated that the rescoring process could be optimized. In conclusion, neither of the mass spectrometric information (xiSCORE) nor the RT information (SVM score) seem to reveal all true CSMs.

As a combination of both approaches should yield better results than either alone, we combined the SVM score with the xiSCORE. We evaluated the impact of rescoring CSMs on the number and quality of identified PPIs, as PPIs are typically the objective of large-scale cross-linking MS experiments.

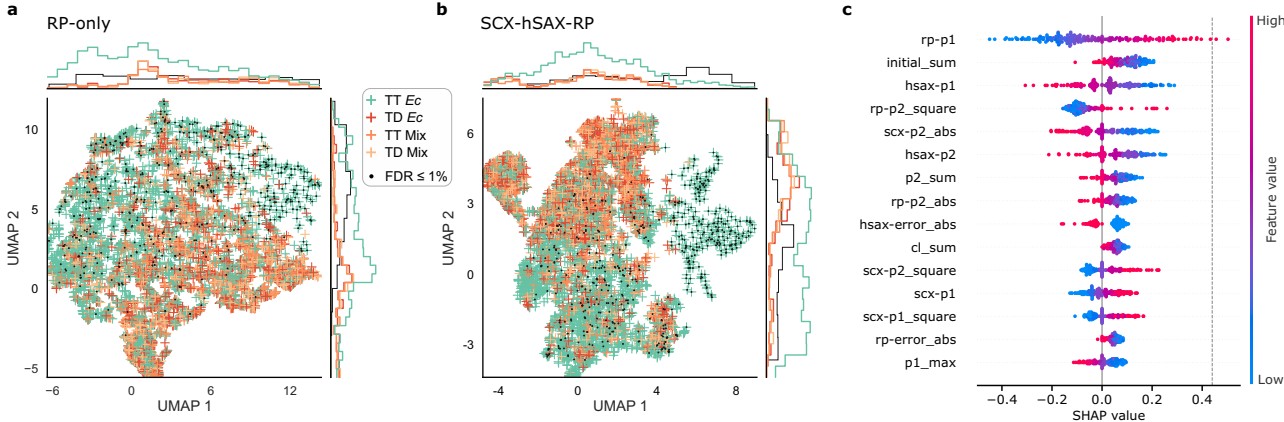

**Fig. 3 Visualization of RT features. a** xiRT-based features from reversed-phase chromatography (RP) dimension only (13 features) after dimensionality reduction with Uniform Manifold Approximation and Projection (UMAP). **b** xiRT-based feature from SCX-hSAX-RP (strong-cation exchange – hydrophilic strong-anion exchange – reversed-phase chromatography) dimensions (43 features) after dimensionality reduction with UMAP. Input data for **a** and **b** were CSMs of heteromeric links in the proteome-wide crosslinking dataset (*Ec* = *E. coli*; green = TT; red = TD, Mix = match between *E. coli* and human peptides; orange = TT; peach =TD), filtered to 50% CSM-FDR. Identifications passing 1% CSM-FDR are highlighted. Decoy-decoy identifications are not shown. **c** SHAP analysis of RT feature importance for CSM-rescoring (using a linear SVM) including SCX, hSAX and RP features (Supplementary Table 4). Each dot represents a previously identified CSM from 200 randomly chosen TTs that were excluded from training (i.e., CSM-FDR > 1%). The background data set consists of 100 TT and TD CSMs each. Dashed line indicates the base value for a prediction based on the background data alone (0.44). Source data are provided as a Source Data file.

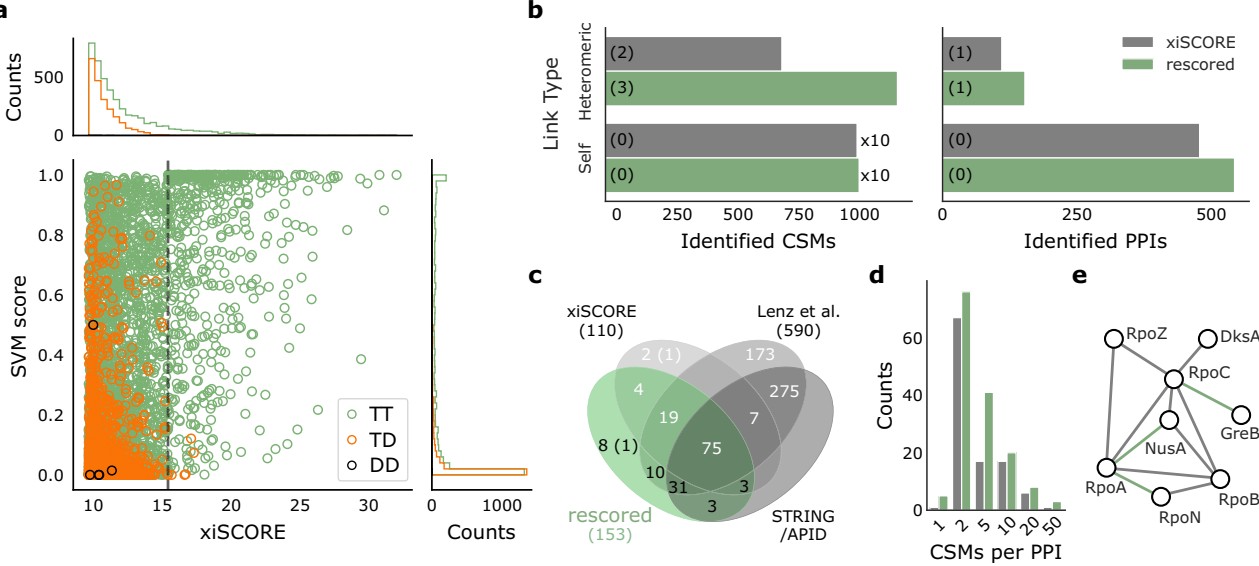

**Fig. 4 Incorporation of RT prediction to CSM-scoring increases crosslink identification. a** Score distributions of heteromeric CSMs based on mass spectrometric information (xiSCORE) and retention time features (SVM score). The dashed line indicates the xiSCORE-based CSM-FDR threshold of 1%. **b** Increase in the identification of TT-CSMs and PPIs at constant FDR. Numbers in brackets indicate identifications involving a human protein. **c** Overlap of observed PPIs (at 1% heteromeric PPI-FDR) to external references. Numbers in the Venn diagram represent the identified PPIs among *E. coli* proteins or PPIs involving human proteins (in brackets). Black numbers highlight the added benefit from combining xiSCORE with xiRT's SVM score for PPI identification. **d** Distribution of CSMs per PPI before (gray) and after CSM-rescoring (green). **e** Selected subnetwork of the RNA polymerase with PPIs only identified after the rescoring connected in green. Data in **b**–**e** correspond to a 1% PPI-FDR (prefiltered at a 5% CSM-FDR). Source data are provided as a Source Data file.

Heteromeric CSMs increased 1.7-fold and heteromeric PPIs increased 1.4-fold (Fig. 4b). Self-links increased only marginally in agreement with their smaller search space and accordingly lower random match frequency. Essentially, nearly all self-links were identified exhaustively based on mass spectrometric data alone. In contrast, RT information substantially improved the identification of heteromeric CSMs. Further gains might be possible by directly combining RT features with mass spectrometric features (and possibly also other) for supervised scoring.

Likely, the benefits of RT predictions for the rescoring depend on the data set and applied chromatographic separations. On the *E. coli* data, we, therefore, performed additional analyses where we limited the rescoring to only use a subset of the chromatographic dimensions (Supplementary Table 5). The number of identified CSMs for heteromeric links increased from 724 in the reference to 902 (RP only), 977 (SCX-RP), 1092 (hSAX-RP), and 1199 (SCX-hSAX-RP). Likewise, PPIs increased from 109 to 135, 131, 157, 152, respectively (Supplementary Table 5). As observed

above, gains can be expected from each chromatographic dimension. When having to choose one ion chromatography, the hSAX dimension seemed more useful than the SCX dimension which could arise from the better prediction performance or more complex separation mechanisms. Importantly, even using RP RT alone already led to a marked gain in heteromeric PPIs (see also next section).

To systematically evaluate the additionally identified PPIs from all three RT dimensions, we compared them to the originally identified PPIs based exclusively on xiSCORE. In addition, the STRING/APID databases and a set of PPIs from a larger study[16] served as extra references for validation. Almost all PPIs found in the original dataset by xiSCORE were also contained in the rescored data set (91%). 85% of the newly identified PPIs were either found in the data set from Lenz et al., in STRING/APID or both. Among the eight PPIs unique to the rescored data set, only one involved a human protein from the entrapment database (Fig. 4c), which we could manually resolve and match to *E. coli* (Supplementary Table 6). The remaining seven PPIs might constitute genuine PPIs. Note that the overall percentage of PPIs involving human proteins was reduced by rescoring. Since all human target proteins were included in the positive training data, this is an important indicator of a well-behaved model. Deepening trust further, almost all novel PPIs were identified with multiple CSMs (Fig. 4d). Finally, we selected the subnetwork of the RNA polymerase to investigate the additionally identified PPIs in a well-characterized interaction landscape (Fig. 4e). Indeed, all interactions added by RT-based rescoring were already reported in APID. In summary, all our evidence points at the successful complementation of MS information by RT, at least for a proteome-wide crosslinking analysis. It remained to be seen, however, if this could also be leveraged in more routine multiprotein complex analyses.

**Multiprotein complex studies also benefit from the RT prediction**. Many cross-linking MS studies investigate multiprotein complexes and rely on only a few chromatographic dimensions. We, therefore, evaluated the benefit of predicted RTs for the analysis of the FA-complex, an eight-membered multiprotein complex that was crosslinked using BS3. Here, the search engine score was supplemented exclusively with RP RT predictions during the rescoring. By using transfer learning, the small number of CSMs (692 unique CSMs, without considering charge states) found in this multiprotein complex analysis were sufficient to achieve accurate RP predictions (Supplementary Fig. 10). The resulting crosslinks at 1% residue-pair FDR (lower levels set to 5%) showed an increase of 36 (+10%) self- and 53 (+70%) heteromeric residue-pairs. Importantly, the rescored links showed no indication of increased hits to the entrapment database (Fig. 5a) indicating that no overfitting occurred during the rescoring. At the same time, heteromeric PPIs already identified before rescoring received additional support. For example, the number and sequence coverage of links increased between FAAP100 (100) and FANCB (B), FANCA (A) and FANCB, and FANCA and FANCG (G). Overall, the heteromeric links increased 1.7-fold with an even higher proportional increase in "verified" links, i.e., fitting the available structure, by 1.9-fold (Fig. 5b). The derived distance distribution of newly identified links is dissimilar from a random distribution and shows no indications of reduced quality (Fig. 5c). Applying this "structural validation" on its own might be optimistic[49], however, in summary, our rigorous quality control ensures trustworthy results. It is currently unclear how far even smaller data sets could benefit from xiRT. Generally, to improve prediction performance, pre-training on larger data sets will lead to better generalization

abilities of the predictor. Subsequently, also smaller data sets can be used for accurate RT prediction. To additionally benefit from sample-specific information, increasing the cross-validation splits will utilize larger parts of the data during training. In any case, our successful analysis of a multiprotein complex supplemented with only RP features highlights the broad applicability of xiRT.

Using a Siamese network architecture, we succeeded in bringing RT prediction into the Crosslinking MS field, independent of separation setup and search software. Our open-source application xiRT introduces the concept of multi-task learning to achieve multi-dimensional chromatographic retention time prediction and may use any peptide sequence-dependent measure including for example collision cross-section or isoelectric point. The black-box character of the neural network was reduced by means of interpretable machine learning that revealed individual amino acid contributions towards the separation behavior. The RT predictions—even when using only the RP dimension—complement mass spectrometric information to enhance the identification of heteromeric crosslinks in multiprotein complex and proteome-wide studies. Overfitting does not account for this gain as known false target matches from an entrapment database did not increase. Leveraging additional information sources may help to address the mass-spectrometric identification challenge of heteromeric crosslinks.

## Methods

**Sample preparation and multidimensional fractionation**. Biomass was produced from a single clone of *Escherichia coli* K12 strain (BW25113 purchased from DSMZ, Germany; https://www.dsmz.de/) by fermentation in a Biostat A plus bioreactor (Sartorius, Göttingen, Germany) in LB medium with 0.5% (w/v) glucose at 37 °C while monitoring and adjusting pH and dissolved oxygen by the addition of sodium hydroxide/phosphoric acid or stir speed control, respectively. When the culture grew to an optical density$_{600}$ of 10 it was harvested by centrifugation at 5000×*g*, 4 °C for 15 min, then washed with 1× PBS, aliquoted, snap-frozen in liquid nitrogen, and stored at −80 °C. Cell pellets were resuspended in lysis buffer (50 mM Hepes pH 7.2 at RT, 50 mM KCl, 10 mM NaCl, 1.5 mM MgCl$_2$, 5% (v/v) glycerol, 1 mM dithiothreitol (DTT), spatula tip of chicken egg white lysozyme (Sigma, St. Louis, MO, USA)) and lysed by sonication. Prior to sonication, cOmplete EDTA-free protease-inhibitors (Roche, Basel, Switzerland) were added according to the manufacturer's instructions. Then, Benzonase (Merck, Darmstadt, Germany) was added and the lysate cleared from cellular debris by centrifugation for 15 min at 4 °C and 15.000×*g*. Fresh DTT was supplied to 2 mM. The obtained supernatant was treated further by ultracentrifugation using a 70 Ti fixed-angle rotor for 1 h at 106,000×*g* and 4 °C. Subsequently, the protein solution was concentrated using Amicon spin filters (15 kDa molecular weight cut-off; Merck, Darmstadt, Germany) to reach a total protein concentration of 10 mg/ml, as judged by microBCA assay (ThermoFisher Scientific, Waltham, MA, USA) and aggregates removed by centrifugation for 5 min at 16,900×*g* and 4 °C. Then, 2 mg of this soluble high molecular weight proteome was separated on a BioSep SEC-S4000 column (600 × 7.8 mm, pore size 500 Å, particle size 5 μm, Phenomenex, CA, USA) at 200 μl/min flow rate and 4 °C with fraction collection of 200 μl over the separation range from ~3 MDa to 150 kDa (as judged by Gel filtration calibration kit (HMW), GE Healthcare) to give 44 fractions. The proteins of each fraction were crosslinked using 0.75 mM disuccinimidyl suberate (DSS; Sigma, St. Louis, MO, USA). The cross-linked samples were pooled and precipitated using acetone. Upon resuspending in 6 M urea, 2 M thiourea, 100 mM ammonium bicarbonate (ABC), the samples were derivatized by incubating 30 minutes at room temperature with 10 mM dithiothreitol followed by 20 mM iodoacetamide in the dark. Proteolysis was accomplished using LysC protease (1:100 protease-to-substrate mass ratio; Pierce Biotechnology, Rockford, IL, USA) for 4.5 h at 37 °C, followed by 1:5 dilution with 100 mM ABC and additional digestion with and Trypsin (1:25 protease-to-substrate mass ratio; Pierce Biotechnology, Rockford, IL, USA). Digestions were quenched by adding trifluoroacetic acid (TFA) and cleaned up using Stage-tips. The sample was fractionated in the first dimension on a Poly-Sulfoethyl A strong cation exchange chromatography (SCX) column (100 × 2.1 mm, 300 Å, 3 μm) equipped with a guard column of identical stationary phase (10 × 2.0 mm) (PolyLC, Columbia, MD, USA) running at 0.2 ml/min on an Äkta pure system (GE Healthcare, Chicago, IL, USA) at 21 °C. Mobile phase A was 10 mM monopotassium phosphate pH 3.0, 30% acetonitrile; mobile phase B additionally contained 1 M potassium chloride (KCl). About 0.4 mg peptides dissolved in mobile phase A were loaded and eluted isocratically over 2 min, followed by an exponential gradient up to 700 mM KCl with the following steps: 12 min to 12.7%, followed by 1-min steps to 14.5, 16.3, 18.8, 23.0, 30.0, 40.0, 70.0% B. We collected nine high-salt fractions of 0.2 ml size during several replica SCX runs. Identical fractions were pooled and desalted using Stage-tips followed by separation in the

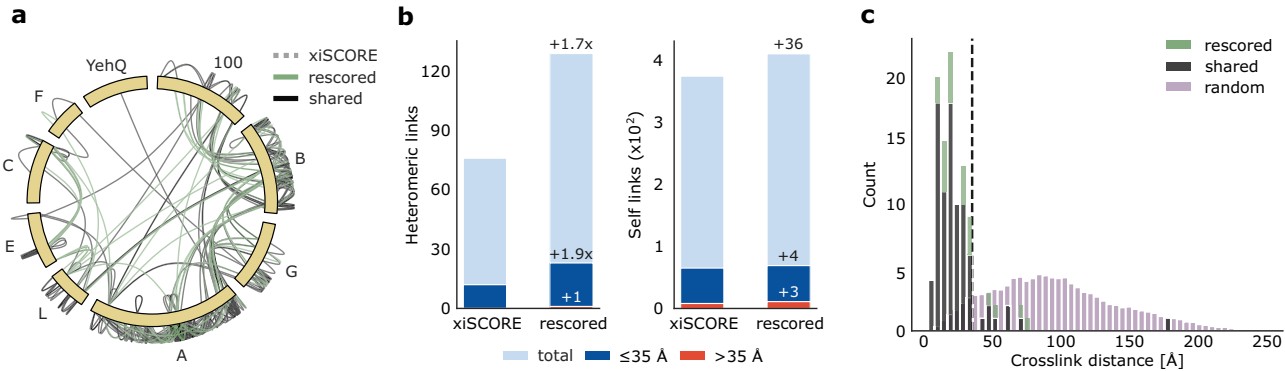

**Fig. 5 Benefit of RT prediction for multiprotein complex crosslink analysis. a** Crosslink network from the Fanconi anemia complex analysis, shown in the circular view. Unique residue pairs from xiSCORE (gray), after rescoring (green), and shared (black) between these analyses are depicted (1% residue-pair FDR). Proteins associated to the Fanconi anemia core complex are indicated with their gene name suffix. The *E. coli* protein YehQ represents a match from the entrapment database. **b** Quantitative assessment of residue-pairs with and without rescoring, and including calculated distances in the model (all, light blue; ≤35 Å, blue; >35 Å red). **c** Distribution of crosslink distances from identified residue-pairs (*n* = 105) following rescoring (green), shared (black) between rescoring and xiSCORE (since no crosslinks unique to xiSCORE), and theoretically possible residue-pairs (random) that could be mapped to the model. Source data are provided as a Source Data file.

second chromatographic dimension by hydrophilic strong anion exchange chromatography (hSAX). Here, we used a Dionex IonPac AS-24 hSAX column (250 × 2.0 mm) with an AG-24 guard column (Thermo Fisher Scientific, Dreieich, Germany) running at 0.15 ml/min on an Äkta pure system (see above) and at 15 °C. Mobile phases A and B were 20 mM Tris*HCl pH 8.0 with B additionally containing 1 M sodium chloride. Samples were loaded in mobile phase A and separated under isocratic conditions for 3 min, followed by elution using an exponential gradient: 1.8, 3.5, 5.3, 7.1, 9.1, 11.2, 13.5, 16.3, 19.7, 24.1, 30.2, 38.8, 51.5, 70.6, 100% B, each step lasting for one minute. Fractions of 0.15 ml size were collected along the gradient. Ten pools were prepared (fractions 3-6/7-14/15-17/18-19/20-21/22-23/24-25/26-27/29-29/30-35) and desalted using Stage-tips.

**LC-MS for crosslink identification.** Analysis of crosslinked peptides by LC-MS was conducted on a Q Exactive HF mass spectrometer (ThermoFisher Scientific, Bremen, Germany) coupled to an Ultimate 3000 RSLC nano system (Dionex, Thermo Fisher Scientific, Sunnyvale, USA), operated under Tune 2.11, SII for Xcalibur 1.5 and Xcalibur 4.2. Solvents A and B were 0.1% (v/v) formic acid and 80% (v/v) acetonitrile, 0.1% (v/v) formic acid, respectively. Peptide fractions were dissolved and loaded in 1.6% acetonitrile, 0.1% formic acid onto an Easy-Spray column (C18, 50 cm, 75 µm ID, 2 µm particle size, 100 Å pore size) operated at 300 nl/min flow and 45 °C. Peptide elution used the following gradient: 2 to 7.5% buffer B within 5 min, from 7.5 to 42.5% over 80 min, to 50% B over 2.5 min, and then to 95% buffer B within 2.5 min and flushed for another 5 min before re-equilibration at 2% B. Survey scans were acquired at a resolution of 120,000, automated gain control of $3*10^6$, maximum injection time of 50 ms while scanning from 400–1450 *m/z* in profile mode. The top 10 intense precursor ions with *z* = 3-6 and passing the peptide match filter (preferred) were isolated using a 1.4 *m/z* window and fragmented by higher-energy collisional dissociation using stepped normalized collision energies of 24, 30, and 36. Fragment ion scans were recorded at a resolution of 60,000, with automated gain control set to $5*10^4$, maximum injection time of 120 ms, underfill ratio of 1%, and scanning from 200–2000 *m/z*. Dynamic exclusion for previously fragmented precursors and their isotopes was enabled for 30 s. To minimize the non-covalent gas-phase association of peptides, in-source-CID was enabled at 15 eV[36]. Each LC-MS run lasted for 120 min.

**Spectra and peptide spectrum match processing.** All raw spectra were converted to Mascot generic format (MGF) using msConvert[50] (3.0.20175.cbf82d022). The database search with Comet[51] (v. 2019010) was done with the following settings: peptide mass tolerance 3 ppm; isotope_error 3; fragment bin 0.02; fragment offset 0.0; decoy_search 1; fixed modification on C (carbamidomethylation, +57.021 Da); variable modifications on M (oxidation, +15.99 Da). False discovery rate (FDR) estimation was performed for each acquisition. First, the highest-scoring PSM for a modified peptide sequence was selected, then the FDR was computed based on Comet's *e*-value. Spectra were searched using xiSEARCH (v. 1.6.753)[12], after recalibration of precursor and fragment *m/z* values, with the following settings: precursor tolerance, 3 ppm; fragment tolerance, 5 ppm; missed cleavages, 2; minimum peptide length, 7; variable modifications: oxidation on M, mono-links for linear peptides on K, S, T, Y, fixed modifications: carbamidomethylated C. The specificity of the crosslinker DSS was configured to link K, S, T, Y, and the protein N terminus with a mass of 138.06807 Da. The searches were run with the workflow system snakemake[53]. The FDR on CSM-level was defined as FDR = TD − DD/TT[40], where TD indicates the number of target-decoy matches, DD the number of decoy–decoy matches, and TT the

number of target-target matches. Crosslinked peptide spectrum matches (CSMs) with non-consecutive peptide sequences were kept for processing[48]. PPI level FDR computation was done using xiFDR[40] (v. 2.1.3 and 2.1.5 for writing mzIdentML) to an estimated PPI-FDR of 1%, disabling the boosting and filtering options. CSM, peptide, and residue-level FDR were fixed at 5%, protein group FDR was set to 100%. FDR estimations for self and heteromeric links were done separately. In xiFDR a unique CSM is defined as a combination of the two peptide sequences including modifications, link sites, and precursor charge state. For the assessment of identified CSMs an entrapment database (described in the next section), as well as decoy identifications, were used on both, CSM and PPI levels. PPI results were also compared against the APID[42] and STRING[41] databases (v11, minimal combined confidence of 0.15).

**Database creation.** The database of potentially true crosslinks was defined as *Escherichia coli* proteome (reviewed entries from Uniprot release 2019-08). This database was filtered further to proteins identified with at least a single linear peptide at a *q*-value[54] threshold of 0.01, $q(t) = \min_{s \leq t} FDR(s)$, with the threshold *t* and score *s*. This resulted in 2850 proteins. In addition to the FDR estimation through a decoy database, we used an entrapment database. The proteins from the entrapment database represent the search space of false-positive CSMs independent of *E. coli* decoys and were sampled from human proteins (UP000005640, retrieved 2019-05). *E. coli* decoys might fail in this task after machine learning if overfitting should have taken place. So, entrapment targets allow control for overfitting. For this, human target peptides were treated as targets and human decoy peptides as decoys. To avoid complications through false spectrum matches due to homology, we used blastp[55] (BLAST 2.9.0+, blastp-short mode, word size 2, *e*-value cutoff 100) and aligned all *E. coli* tryptic peptides (1 missed cleavage, maximum length 100) to the human reference. All proteins that showed peptide alignments with a sequence identity of 100% were removed from the human database. Only the remaining 9990 sequences were used as candidates in the entrapment database. For each of the 2850 *E. coli* proteins, a human protein was added to the database. To reduce search space biases from protein length and thus different number of peptides for the two organisms, we followed a special sampling strategy. The human proteins were selected by a greedy nearest neighbor approach based on the K/R counts and the sequence length. The final number of proteins in the combined database (*E. coli* and human) was 5700 (2850*2).

**Fanconi anemia monoubiquitin ligase complex data processing.** The publicly available raw files from an analysis of the BS3-crosslinked Fanconi anemia monoubiquitin ligase complex[56] (FA-Complex) were downloaded from PRIDE together with the original FASTA file (PXD014282). The raw files were processed as described for the *E. coli* data (*m/z* recalibration and searched with xiSEARCH), followed by an initial 80% CSM-FDR filter for further processing. Due to the much smaller FASTA database (8 proteins), the entrapment database was constructed more conservative than for the proteome-wide *E. coli* experiment, i.e., for each of the target proteins, the amino acid composition was used to retrieve the nearest neighbor in an *E. coli* database. The FDR settings to evaluate the rescoring were set to 5% CSM- and peptide-pair level FDR, 1% residue-pair- and 100% PPI-FDR using xiFDR without boosting or additional filters. The resulting links were visualized (circular view) and mapped to an available 3D structure (final refinement model "sm.pdb")[57,58] using xiVIEW[59]. To ease the comparison of identified and random distances, a random Euclidean distance distribution was derived in three steps: first, all possible cross-linkable residue-pair distances in the 3D

structure were computed. Second, 300 random "bootstrap" samples with n distances were drawn (n = the number of identified residue-pairs at a given FDR) and third, the mean per distance bin was computed across all 300 samples.

**xiRT—3D Retention Time Prediction**. The machine learning workflow was implemented in python (v. >3.7) and is freely available from https://github.com/Rappsilber-Laboratory/xiRT. xiRT is the successor of DePART[29], which was developed for the retention time (RT) prediction of hSAX fractionated peptides based on pre-computed features. xiRT makes use of modern neural network architectures and does not require feature engineering. We used the popular python packages sklearn[60] (0.24.1) and TensorFlow[61] (v. 1.15 and >2) for processing (Supplementary Note 1 for more details). xiRT consists of five components (Fig. 1d and Supplementary Fig. 1, Supplementary Note 1): (1) The input for xiRT are amino acid sequences with arbitrary modifications in text format (e.g., Mox for oxidized Methionine). xiRT uses a similar architecture for linear and crosslinked peptide RT prediction. Before the sequences can be used as input for the network, the sequences are label encoded by replacing every amino acid by an integer and further 0-padded to guarantee that all input sequences have the same length. Modified amino acids, as well as crosslinked residues, are encoded differently than their unmodified counterparts. (2) The padded sequences were then forwarded into an embedding layer that was trained to find a continuous vector representation for the input. (3) To account for the sequential structure of the input sequences, a recurrent layer was used (either GRU or LSTM). Optionally, the GRU/LSTM layers were followed by batch normalization layers. For cross-linked peptide input, the respective outputs from the recurrent layers were then combined through an additive layer (default setting). (4) Task-wise subnetworks were added for hSAX, SCX, and RP retention time prediction. All three subnetworks had the same architecture: three fully connected layers, with dropout and batch normalization layers between them. The shape of the subnetworks is pyramid-like, i.e., the size of the layers decreased with network depth. (5) Each subnetwork had its own activation function. For the RP prediction, a linear activation function was used and mean squared error (MSE) as loss function. For the prediction of SCX and hSAX fractions, we followed a different approach. The fraction variables were encoded for ordinal regression in neural networks[62]. For example, in a three-fraction setup, the fractions ($f$) were encoded as $f_1 = [0, 0, 0], f_2 = [1, 0, 0]$ and $f_3 = [1, 1, 0]$. Subsequently, we chose sigmoid activation functions for the prediction layers and defined binary cross-entropy (BC) as loss function. To convert predictions from the neural network back to fractions, the index of the first entry with a predicted probability of <0.5 was chosen as the predicted fraction. The overall loss was computed by a weighted sum of the $MSE_{RP}$, $BC_{SCX}$, and $BC_{hSAX}$. The weight parameters are only necessary when xiRT is used to predict multiple RT dimensions at the same time (multi-task). To predict a single dimension (single-task, e.g., RP only), the weight can be set to 1. The number of neurons, dropout rate, intermediate activation functions, the weights for the combined loss, number of epochs, and other parameters in xiRT were optimized on linear peptide identification data. Reasonable default values are provided within the xiRT package. For optimal performance, further optimization might be necessary for a given task.

**Cross-validation and prediction strategy**. Cross-validation (CV) is a technique to estimate the generalization ability of a machine learning predictor[63] and is often used for hyper-parameter optimization. We performed a 3-fold CV for the hyper-parameter optimization on the linear peptide identification data from xiSEARCH, excluding all identifications to the entrapment database (Supplementary Note 2 and Supplementary Fig. 2 for details). We defined a coarse grid of parameters (Supplementary Table 1) and chose the best performing parameters based on the average total (unweighted) loss, $R^2_{RP}$ and accuracy across the CV folds. Further, we define the relaxed accuracy (racc) to measure how many predictions show a lower prediction error than |1| fraction. We then repeated the process with an adapted set of parameters (Supplementary Table 2). In addition to the standard CV strategy, we used a small adjustment: per default, in $k$-fold cross-validation, the training split consists of $k - 1$ parts of the data (folds) and a single testing fold. However, we additionally used a fraction (10%) from the training folds as extra validation set during training. The validation set was used to select the best performing classifier over all epochs. The model assessment was strictly limited to the testing folds. This separation into training, validation, and testing was also used for the semi-supervised learning and prediction of RTs, i.e., when xiRT was used to generate features to rescore CSMs previously identified from mass spectrometric information. In this scenario, the CV strategy was employed to avoid the training and prediction on the same set of CSMs. In xiRT, a unique CSM is defined as a combination of the two peptide sequences, ignoring link sites and precursor charge.

**Supervised peptide spectrum match rescoring**. To assess the benefits of RT predictions, we used a semi-supervised support vector (SVM) machine model. The implementation is based on the python package scikit-learn[60] in which optimal parameters are determined via cross-validation. The input features were based on the initial search score (for FA-complex only) and differences between predicted and observed RTs. For each cross-linked peptide, three predictions were made per chromatographic dimension: for the crosslinked peptide, for the alpha peptide, and the beta peptide. Additional features were engineered depending on the number of

chromatographic dimensions and included the summed, absolute, or squared values of the initial features (Supplementary Table 3 for all features). For example, for three RT dimensions, the total number of features was 43. The data for the training included all CSMs that passed the 1% CSM-FDR cutoff (self, heteromeric/TT, TD, DDs) and TD/DD identifications that did not pass this cutoff. TTs were labeled as positive training examples, TD and DDs (DXs) were labeled as negative training examples.

To stratify the $k$-folds during CV, the CSMs were binned into $k$ xiSCORE percentiles. Afterward, they were sampled such that each score range was equally represented across all CV folds. When the positive class was limited to the TT identifications at 1% CSM-FDR, the number of negative observations was usually larger than the number of positive observations. To circumvent this, for each CV split, a synthetic minority over-sampling technique (SMOTE)[64] was used to generate a balanced number of positive and negative training samples (here only used for the FA-complex data). SMOTE was applied within each CV fold to avoid information leakage. A 3-fold CV was performed for the rescoring. In each iteration during the CV, two folds were used for the training of the classifier, and the third fold was used to compute an SVM score. During this CV step, a total of three classifiers were trained. The scores for all TT-CSMs that did not pass the initial FDR cutoff were computed by averaging the score predictions from the three predictors. For all CSMs passing the initial FDR cutoff, rescoring was performed when the CSM occurred in the test set during the CV. The final score was defined as: $xi_{rescored} = xi_{SCORE} + xi_{SCORE} \times SVM_{score}$, where $SVM_{score}$ was the output from the SVM classifier and $xi_{SCORE}$ the initial search engine score.

**Feature analysis**. The KernelExplainer from SHAP[65] (Shapley Additive exPlanations, v.0.36.0) was used to analyze the importance of features derived from the SVM classifier. SHAP estimates the importance of a feature by setting its value to "missing" for an observation in the testing set while monitoring the prediction outcome. We used a background distribution of 200 samples (100 TT, 100 TD) from the training data to simulate the "missing" status for a feature. SHAP values were then computed for 200 randomly selected TT (predicted to be TT) that were not used during the SVM training. SHAP values allow to directly estimate the contributions of individual features towards a prediction, i.e., the expected value plus the SHAP values for a single CSM sums to the predicted outcome. For a selected CSM, a positive SHAP value contributes towards a true match prediction. For the interpretability analysis (SHAP) of the learned features in xiRT, the DeepExplainer was used (Supplementary Note 3).

In addition, we performed dimensionality reduction using UMAP[66] on the RT feature space for visualization purposes (excluding the search engine score). UMAP was run with default parameters (n_neighbors = 15, min_dist = 0.1) on the standardized feature values. The list of used features for the multi-task learning setup is available in Supplementary Table 3.

**Statistical analysis**. Significance tests were computed using a two-sided independent $t$-test with Bonferroni correction. The significance level $\alpha$ was set to 5%.

**Reporting summary**. Further information on research design is available in the Nature Research Reporting Summary linked to this article.

## Data availability
The mass spectrometry proteomics data have been deposited to the ProteomeXchange Consortium (http://proteomecentral.proteomexchange.org) via the jPOST partner repository[67] with the data set identifier PXD020407 and at https://doi.org/10.6019/PXD020407. Raw data of the FA-Complex are available via the previously published PRIDE identifier PXD014282. Additional files and intermediate results are available via Zenodo at https://doi.org/10.5281/zenodo.4270323. PPI data were retrieved from STRING (https://string-db.org/, v11) and APID (http://cicblade.dep.usal.es:8080/APID/init.action, downloaded 09/2019). Source data are provided with this paper.

## Code availability
The developed python package is available on the python package index, on GitHub (https://github.com/Rappsilber-Laboratory/xiRT) and via Zenodo (https://doi.org/10.5281/zenodo.4270323).

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

## Acknowledgements

We thank Edward Rullmann, Andrea Graziadei, and Francis J. O'Reilly for the critical reading of the manuscript, and Jakub Bartoszewicz (RKI / HPI) for fruitful discussions. We are grateful to Tabea Schütze for help with fermenting *E. coli*. This work was supported by NVIDIA with hardware from the grant "Artificial Intelligence for Deep Structural Proteomics", by the Wellcome Trust through a Senior Research Fellowship to J.R. (103139) and by the Deutsche Forschungsgemeinschaft (DFG, German Research Foundation) under Germany's Excellence Strategy - EXC 2008 - 390540038 – UniSysCat, and by grant no. 392923329/GRK2473. The Wellcome Centre for Cell Biology is supported by core funding from the Wellcome Trust (203149).

## Author contributions

Study design: S.H.G., L.R.S., and J.R. Software implementation: S.H.G. Sample preparation and mass spectrometry acquisition: L.R.S. and F.W. Data analysis: S.H.G., L.R.S., and J.R. All authors critically evaluated and approved the manuscript.

## Funding

## Competing interests

The authors declare no competing interests.
