## [Peer Review File · Nature Communications]

REVIEWER COMMENTS

Reviewer #2 (Remarks to the Author):

In this manuscript, the authors present a novel means to incorporate retention time information into cross-linking mass spectrometry data analyses. The authors integrate this through a new learning model and python package termed “xiRT” which complements other works in the “xi” lineup (xiView, xiNET, xiSearch, etc.). They then go on to show that the use of xiRT enables a “...2-fold increase in protein-protein interactions detected...”, which is an impressive improvement. The utilization of the retention time dimension in LC-MS/MS analyses is an interesting one, but also one that has in part been previously devised by this group (Chen & Rappsilber, Nat Protocols [2019]). The main advance here seems to be the use of the new learning model to estimate retention times and use this estimation to filter out spurious cross-linked peptide identifications. The predominant effect being improved retention of heteromeric linkages. The implementation of this learning model is interesting, though the widespread applicability seems potentially limited. I commend the authors on making the data and the code available. My main concerns surround the extensibility of this platform to other workflows (there are many in crosslinking MS) and how the model/entrapment database were built.

Based on the above and the comments below, I believe this paper may be of interest to the Nature Communications audience with the requested revisions to include information that is essential for understanding the utility of the project.

Major Concerns

1. Entrapment database. The authors spend a great deal of effort describing the entrapment database used to establish their false discovery rate estimates and test the xiRT model. The database is made up of two filtered proteomes (human and E. coli). The authors minimize direct overlap of the two cohorts of 2850 proteins (i.e. BLASTp based filter) but do little to show the properties of the resulting peptides are consistent within the databases. Plots describing these properties should be included at the very least, but I encourage the authors to thoroughly explore this, and if they have already done so (which is assumed based on the mention of using “...a greedy nearest neighbor approach ...”[p6, ln137]) it is essential that they report these data to properly assess the manuscript.

a. Amino acid composition, hydrophobicity, and peptide length would be the most important for comparison as these will have a direct effect on retention time. The authors make minor mention of this (p13-14, ln329-330), but offer very little to demonstrate the bold claim that “...physicochemical differences are therefore not explaining the observed effect.”

2. Decoy annotation. Could the authors clarify if human proteins were considered “decoys” for the purposes of FDR estimation? Adding this to the “Spectra & Peptide Spectrum Match Processing” section would be helpful.
3. Database availability. The final database should be included with the manuscript. There are 6 FASTA files in the JPost project, none of which are clearly labeled as the “entrapment database”.
4. Entrapment comparison. Did the authors try to use and compare the RT diff with Percolator or PeptideProphet as in their previous work (Mendes et al MSB [2019])?
5. Model performance. Do the authors have a sense of why MSE for the SCX and SAX differ by 2- to 3-fold (Fig 2, S3) for CSMs and linear peptides at 40+ epochs? The number of fractions were similar as well as the CSM count profiles (Figure S1). This extends to: why was the SAX accuracy higher than the SCX accuracy?
6. Model dimensionality. Are the SCX and SAX subnetworks required for the benefit of xiRT, or can these be eliminated for more generalizability to other workflows? Can the authors provide a comparison of the learned model with only the RP subnetwork? This would be important as most groups do not use the fractionation strategy demonstrated in this work (SEC + SAX + SCX + RP).
7. Fractions. Do the authors estimate that their modeling would perform better or worse with larger numbers of fractions or with the removal of either the SAX or SCX fractionation?
8. Utility and Extensibility. The authors spend extensive time determining the utility for a single workflow to investigate an E. coli PPI network. How extensible do they believe this platform will be for the general XL-MS user? Can the authors describe how this software could be used with different workflows beyond xiSearch/xiRescore?
9. Expanding Usage. How do the authors envision extending this to human PPI network investigations, e.g. for building an entrapment database or estimating RTs? Including information on this in the Discussion/Conclusions would aid readers’ understanding of the utility of this technique outside of the demonstrated use-case.
10. Heteromeric linkages. Are the authors concerned that the main benefit between xiScore and xiRescore in Figure 4b for CSMs and PPIs is for heteromeric linkages (the claimed 2.1-fold improvement)? As the author’s know (Lenz et al. preprint), these linkages are more likely to contain decoy matches compared to self-linkages. The Lenz et al. data also appears to be filtered to a 5% FDR rather than the 1% FDR for xiRescore, though it is used for validation. Furthermore, as there are fewer heteromeric human PPIs than CSMs, did the authors observe multiple CSMS within the same human protein pairs? What do they make of these consistent, wrong “interactions”?
11. RT Comparisons. Other groups have used RT for crosslinked peptide matching (e.g. most recently Steigenberger et al MCP [2020]). The authors should provide comparison of whether the 2-fold improvement can be gained through simpler methods – which would be more widely adaptable – to utilize retention time in their analyses.
12. Structure-based FDR. Owing to the large collection of E. coli protein structures, did the authors incorporate any kind of structure-based FDR for estimating intraprotein interactions (Mintseris &

Gygi, 2019, PNAS)? There may even be a chance for validation of the heteromeric linkages in well studied complexes.

13. P8, ln 235-238. Can the authors explain why the 4-fold difference was not explained, while the difference observed for the random draw was 3.17-fold?

14. Fold improvement. The Conclusion section highlights at 2.1% improvement in total heteromeric crosslinks, but the overall improvement is much more modest. As noted in point 10, these are also the most likely to result from false positives.

Minor Concerns

I. Figure 1a is misleading in the depiction of the database usage. The implication is that each spectra is searched against full databases and these are processed by BLAST/Comet to generate the final dataset.

II. P1, ln 19. The margin of error comment is a little misleading as it really represents 11% windows around the fractions. So, with 20 fractions, this would be +/- 2 fractions, etc.

III. P1, ln 22. The claim "...2-fold increase in protein-protein interactions detected..." is misleading based on the actual results presented in Figure 4b.

IV. P2, ln 38. There have been several enrichable linkers developed prior to the PhoX linker and should be referenced. The authors should also mention chromatographic enrichment (e.g. SCX), and IMS enrichment strategies (Schnirch et al Anal Chem [2020], Steigenberger et al, MCP [2020]). All these methods are potentially complimentary.

V. P2, ln 46. The claim of "...multi-dimensional chromatography workflows can yield in the order of 10,000 CSMs at 1-5% false discovery rate (FDR)..." should have a reference associated with it or be removed.

VI. The author's consistently use lower case lettering for acronyms (e.g. "blast" in Figure 1a) and the first letter of named programs (e.g. "comet" [p5, ln104]). These should be changed throughout the figures and manuscript to the correct usages.

VII. The authors set up "three major challenges" in the opening paragraph (p2, ln 28-36), but do not mention how their new pipeline solves the issues of low abundance, unequal fragmentation, or combinatorial complexity. Could they provide details on where they think the current work fits into to solving these central issues?

VIII. The "Sample Preparation" section of the Methods makes no mention of reducing the proteins prior to alkylaton. Was this step performed?

Reviewer #3 (Remarks to the Author):

The manuscript by Giese et al. describes a machine learning algorithm for the prediction of crosslinked peptide retention times under multidimensional fractionation techniques. When using the difference between these predictions and the observed retention times additional parameters were produced that improved the number protein-protein interactions observed at various FDR thresholds. These additional interactions shown to be plausible when compared to entrapment interactions, suggest that RT prediction is a valuable tool to improve crosslink search results over search score alone.

The concepts in the paper are interesting, and parallel beneficial efforts that were made in single peptide identification and validation. The authors provide their software tool (xiRT), which appears to be generalizable to workflows outside their lab, and thus a widely useful tool to the community. Considering that crosslinked peptide identification has typically been fraught with false identifications, and that current efforts have largely focused on controlling the error by adopting stricter thresholds, this work perhaps promises to improve analyses through retention of more [correct] identifications following validation. I see this research as having a positive impact on the crosslinking community.

The manuscript does contain several points that need clarification prior to publication:

1. xiRT RT prediction appears to be based on DePART, which is merely referenced in the manuscript. I feel this is a critical component that requires at least a brief overview, even if in the supplementary information.
2. 3D fractionation does not appear to be commonplace to me. Most crosslinking research seems to use simply RP fractionation, and possibly the addition of SCX. Though the online instructions for xiRT possibly indicate it works with RP-only data, the manuscript is not clear on this fact. How much benefit could be expected if simpler fractionation schemes are used (SCX-RP or RP only, for example)? This might be highly relevant considering the RP model struggled with TD matches (see line 322).
3. I found the SHAP analysis difficult to follow. (a) feature interactions were designated as ';' in the supp_info, but with '*' in the main text (and described with ';' in the caption). (b) is the use of '*' to indicate that these interactions are actually each predictor multiplied together and treated as a parameter? (c) how was only 10 features (of 130) chosen to illustrate importance of various features? I find it unlikely to assume that all 130 are similar. (d) how come there appears to be a

large difference in the top 10 features for each SHAP plot provided (1 in main text, 2 in supp_info)? Might expanding the analysis beyond 10 features help identify features common among each analysis, and presumably most important to the model?

4. Machine learning algorithms benefit most from large datasets. The datasets here are large, far larger than many publications (144 acquisitions vs. a dozen or fewer in many publications I've seen). The authors noted the limitation and performed analyses on subsets to model the effect. But I feel this effort stopped short of providing realistic context to the readers. What would someone do if they had only 100 CSMs? Can the authors actually recommend minimum dataset sizes for reliable xiRT performance? And if so, are they still applicable when using fewer features (such as performing the analysis on RP-only data)?

5. I attempted to use xiRT and hit two roadblocks that can probably be fixed by improving the online tutorial. (a) xiRT aborted because libcuda.so.1 could not be found. This is because I don't have an NVIDIA card, and I thought from the documentation that CUDA was optional. If not, this needs to be explicitly stated. (b) xiRT config and setup config are not documented. While I think I could navigate one file correctly, I cannot make sense of the parameters listed in the other. Both of these files need clear documentation and tutorials, not simply a link to YAML.

Reviewer #4 (Remarks to the Author):

xiRT did a good job in retention time (RT) prediction for cross-linked peptides (CX). This is certainly a novel method since there are no RT prediction tools for CXs as I know so far. There are 3 main contributions for xiRT as shown in its abstract: (1) Model CXs by using Siamese Net and using multi-task learning for SCX, hSAX, and RT prediction; (2) It is quite accurate for SCX, hSAX, and RT prediction; (3) Percolator-like rescoring based on predicted SCX, hSAX, and RT features would significantly increase the PPI detections at a proteome-scale (E. coli lysate). This work will be useful for CX-MS analysis.

Here are my comments:

1. It is a great idea to use Siamese Net to model CX problems. Although I do not work on the pLink project, I have a similar idea for predicting "something" of CXs, but this work moved faster than us.

2. Multi-task learning (MTL) is also a good idea. My question is, although MTL can save the training and predicting time, I wondered if single-task learning can achieve a better performance? And what the common knowledge did the model learn within the shared layers? The latter question may be too difficult to answer, therefore, at least, authors should show MTL is necessary for higher predicting accuracies.

3. After the model is designed, it is not hard to build deep neural network models for RT prediction using traditional regression techniques, but it is interesting that xiRT uses ordinal regression models for SCX and hSAX prediction. But there is a concern here: the number of fractions should be fixed for deep ordinal regression network, making it difficult to extend for different fraction numbers? How do the authors consider this problem?

4. At line 109, I suggest to put the sentence "Before the identification with xiSEARCH the masses of precursor and fragment ions were recalibrated." into the front of the xiSEARCH settings, otherwise it will be confused that why fragment tolerance is only 5 ppm.

5. At line 144, the authors said that "The input of xiRT are amino acid sequences with arbitrary modifications", but at the next line, it said "... encoded by replacing every amino acid by an integer", obviously the encoding did not really take "arbitrary" modifications into account, at least I did not find it in this paper.

6. At line 186, xiRT uses rbf kernel for its Percolator-like algorithm. Is it necessary to use a non-linear kernel instead of a linear one? As we know, Percolator uses the linear kernel. Besides, the features (score, RT difference, etc.) used by xiRT may have linear properties. The on-linear kernel tends more easily to be overfitting, especially for some scenarios such as CX search which has fewer training samples.

7. At line 190, I don't understand why there are so many (130) features.

8. From line 191 to 193, does it mean all DXs are negative samples? If so, this description may be too complicated to understand. Otherwise, what did the last sentence suppose to mean?

9. At line 200, xiRT uses a 3-fold CV for rescoring steps. It is reliable, but my question is about extensibility. Most of the structural biologists may only concern about protein-complex level identification instead of proteome-wide level identification, which means there may be not that enough CX-peptides for 3-fold CV for protein-complex identification.

10. Line 233, it is a good idea to use PPI-level evidence to validate CX identifications, but the problems are how to estimate the PPI-level error rate itself? How PPI-error-rate increases as the CSM-FDR increases?

11. Line 236, "Randomly drawing pairs of E. coli proteins" may be a wrong simulation to evaluate the false negatives at 10% or 50% CSM-FDR. As we all know, there are a lot of True-False-linked CXs (i.e. half-correct CXs) at x% CSM-FDR, hence the pairs are not totally random, maybe they are mostly "half-random". I suggest authors should re-consider this simulation.

12. Line 152, it said that xiRT uses an additive layer, but it said: "Multiply-layer was used" in "Siamese Architecture" in Supporting Information S1, it might be a small mistake.

13. Fig. 1d, This network illustration might be not 'siamese' enough, readers cannot get what is the siamese net from this figure if they have not heard about it.

14. Fig. 4b, I have two questions here:

a. What is the PPI FDR? I've never heard about it.

b. For the increment of the xiRescore, the proportion of PPI identifications are significantly larger than that of CSMs, does these additionally identified PPIs are “one-hit-wonder”? Or how to evaluate the quality of additionally identified PPIs?

Dr. Zeng, Wen-Feng

REVIEWER COMMENTS

Reviewer #2 (Remarks to the Author):

In this manuscript, the authors present a novel means to incorporate retention time information into cross-linking mass spectrometry data analyses. The authors integrate this through a new learning model and python package termed “xiRT” which complements other works in the “xi” lineup (xiView, xiNET, xiSearch, etc.). They then go on to show that the use of xiRT enables a “...2-fold increase in protein-protein interactions detected...”, which is an impressive improvement. The utilization of the retention time dimension in LC-MS/MS analyses is an interesting one, but also one that has in part been previously devised by this group (Chen & Rappsilber, Nat Protocols [2019]).

We are pleased that the reviewer recognizes the novel means presented in this manuscript for the usage of retention times for Crosslinking MS. Please note that the manuscript by Chen *et al.* describes a protocol for the quantitation of crosslinked peptides. The published workflow uses **measured** retention time to align MS1 features for their quantitation in Skyline. Chen *et al.* did not touch on retention time **prediction**.

The main advance here seems to be the use of the new learning model to estimate retention times and use this estimation to filter out spurious cross-linked peptide identifications. The predominant effect being improved retention of heteromeric linkages. The implementation of this learning model is interesting, though the widespread applicability seems potentially limited. I commend the authors on making the data and the code available. My main concerns surround the extensibility of this platform to other workflows (there are many in crosslinking MS) and how the model/entrapment database were built.

Based on the above and the comments below, I believe this paper may be of interest to the Nature Communications audience with the requested revisions to include information that is essential for understanding the utility of the project.

Major Concerns

1. Entrapment database. The authors spend a great deal of effort describing the entrapment database used to establish their false discovery rate estimates and test the xiRT model. The database is made up of two filtered proteomes (human and E. coli). The authors minimize direct overlap of the two cohorts of 2850 proteins (i.e. BLASTp based filter) but do little to show the properties of the resulting peptides are consistent within the databases. Plots describing these properties should be included at the very least, but I encourage the authors to thoroughly explore this, and if they have already done so (which is assumed based on the mention of using “...a greedy nearest neighbor approach ...”[p6, ln137]) it is essential that they report these data to properly assess the manuscript.

a. Amino acid composition, hydrophobicity, and peptide length would be the most important for comparison as these will have a direct effect on retention time. The authors make minor mention of this (p13-14, ln329-330), but offer very little to demonstrate the bold claim that “...physicochemical differences are therefore not explaining the observed effect.”

We now show that the peptide length, basic (K/R) / aromatic (F/Y/W) / acidic (D/E) amino acid counts, isoelectric point and GRAVY estimates are similar for the target peptides from *E. coli* and the entrapment-peptides from *H. sapiens*, see Figure S4.

2. Decoy annotation. Could the authors clarify if human proteins were considered “decoys” for the purposes of FDR estimation? Adding this to the “Spectra & Peptide Spectrum Match Processing” section would be helpful.

We added this information to the “Database Creation” section. In short, human targets were treated as targets and human decoys were treated as decoys for FDR estimation.

3. Database availability. The final database should be included with the manuscript. There are 6 FASTA files in the JPost project, none of which are clearly labeled as the “entrapment database”.

The jPOST project contains a “readme.txt” file that explains the databases and naming conventions used in the project. Note that only SwissProt curated protein sequences (“reviewed”) were used. The snippet for the FASTA sequences is pasted below (the final database corresponds to “6”).

FASTA:

- 1) EColi_K12_reviewed_20190828 - E. coli reference proteome
- 2) EColi_K12_reviewed_20190828_4389_cometfilter - 1) minus all proteins not identified in a comet search
- 3) HS_proteome_UP000005640_rev_20190905.fasta - Homo sapiens reference proteome
- 4) HS_proteome_UP000005640_rev_20190905_blastfilter.fasta - 3) minus all proteins that contain peptides that were also found in 1) via a blast search (100% seq. coverage)
- 5) HS_proteome_UP000005640_rev_20190905_blastfilter_nearest.fasta - selection from 4) where each protein was selected based on K/R, peptide length to match the proteins in 2)
- 6) EColi_K12_reviewed_20190828_cbnn_filter.fasta - Combined 2) and 5) (comet-blast-nearest-neighbor_filter).

4. Entrapment comparison. Did the authors try to use and compare the RT diff with Percolator or PeptideProphet as in their previous work (Mendes et al MSB [2019])?

We are aware that Kojak uses PeptideProphet and has used Percolator. We contacted the author of Kojak (Michael Hoopman) and learned that the discriminant score in PeptideProphet and the mixture modeling would need to be implemented for each feature (e.g. RP, SCX, hSAX prediction errors in our case) and are thus substantial work on the PeptideProphet codebase would be necessary to incorporate our features. Therefore, extending PeptideProphet to use RT features is not feasible within this work. However, we hope that our work motivates the author of Kojak/PeptideProphet to consider adding RT features to their workflow. Note that the author of Kojak recommends using PeptideProphet over Percolator (Kojak release updates “I recommend

switching to PeptideProphet in the Trans-Proteomic Pipeline.” <http://www.kojak-ms.org/news/archive2017.html>). The higher error rates when using Percolator have also been shown by an independent group [4].

5. Model performance. Do the authors have a sense of why MSE for the SCX and SAX differ by 2- to 3-fold (Fig 2, S3) for CSMs and linear peptides at 40+ epochs? The number of fractions were similar as well as the CSM count profiles (Figure S1). This extends to: why was the SAX accuracy higher than the SCX accuracy?

1) Linear vs. crosslink performance

The number of epochs that a neural network needs to be trained to achieve good training / validation performance is a function of multiple parameters. Important ones include the number of training samples and thus the number of batches for a given batch-size. For each batch that is used for training, the parameters of the network are updated via gradient descent. With more training data, more batches are needed and thus more adjustments to the initial weights can be performed within a single epoch. Since the linear peptide identifications by far exceed crosslinked peptide identifications, the two figures cannot be quantitatively compared at specific epochs. At the same epoch, the linear peptide network has seen much more data and has performed more updates to the weights. In addition, the combinatorics for linear peptide combinations are not as complex as crosslinked peptide combinations. Therefore, ‘slower learning’ of crosslinked spectrum matches follows our expectation.

2) SCX vs. hSAX

From our experience, crosslinked peptides distribute more equally across the used hSAX gradient, while for SCX they are typically enriched in the later fractions of the gradient. Therefore, the peptides are better separated using hSAX with less redundant peptide pairs across fractions. We also observed this effect when comparing non-unique CSMs (Peptide1, Peptide2, Link Sites, Charge) at 1% FDR. By counting the unique CSMs and redundant CSMs that are found in the same fraction one can estimate the theoretical upper bound on the prediction accuracy (see Figure S8 and Table S4). For hSAX (73%) this upper bound is higher than for SCX (65%) in our data. We have now included these estimations in the manuscript to make this clearer.

6. Model dimensionality. Are the SCX and SAX subnetworks required for the benefit of xiRT, or can these be eliminated for more generalizability to other workflows? Can the authors provide a comparison of the learned model with only the RP subnetwork? This would be important as most groups do not use the fractionation strategy demonstrated in this work (SEC + SAX + SCX + RP).

We are aware that there is no standard regarding the peptide fractionation. Therefore, we developed xiRT for flexible use cases of one or more retention time dimensions. Upon request by reviewer 4 (comment 2), we have added a single-task vs. multi-task comparison to the supplementary material (S6-S7), in which we also probed xiRT by forwarding only RP information. Here, we did not find significant performance differences in retention prediction between the 1-task (RP-only), 2-task (SCX-RP/hSAX-RP) or 3-task (SCX-hSAX-RP) models but reduced runtime.

We then moved on to investigate if RP retention time only can help in identifying crosslinks. We thought of an extreme test, i.e. to look at the ability of RP retention only to separate true and false matches without addition of any mass spectrometric data and added this as new Figure 3a. As one might expect, this does not lead to a full separation using UMAP. Nonetheless, information on RP retention behavior clearly leads to some separation of true and false matches. This improves very much when adding further retention information (hSAX and SCX) in Figure 3b. We would like to emphasize again that this uses no mass spectrometric information. This means that retention time alone carries substantial information that can help distinguish true from false matches.

Finally, to push the RP RT only question even further, we tested xiRT with a much smaller data set from Shakeel *et al.* [3], with 20 acquisitions (BS3 crosslinker used in this data set) for which we only used RP retention data. When we trained xiRT on CSMs filtered to 1% CSM-FDR, the model could not converge and only resulted in an r^2 of 0.49. However, when we pretrained xiRT on the data from this manuscript (DSS-crosslinked *E. coli* proteome) we achieved an average r^2 of 0.91 with weight-adjustment and an r^2 without weight-adjustment 0.37. We added this analysis as new Figure S10. This means that xiRT is applicable to purified complexes in an RP-only workflow and thus generalizes to the currently most widely used application area of crosslinking mass spectrometry (Steigenberger *et al.* 2020 [7]). Notably, we see a 1.7-fold increase in heteromeric links in this analysis at constant FDR.

7. Fractions. Do the authors estimate that their modeling would perform better or worse with larger numbers of fractions or with the removal of either the SAX or SCX fractionation?

We anticipate that a more fine-grained fractionation leads to better prediction performance. However, as shown in response to point 5 of this reviewer, redundant CSMs across fractions will remain a challenge (see Table S4). As stated in our response to the previous question, we have added the single-task vs. multi-task comparison to the supplementary material (S6-S7), in which we did not find significant performance differences between the 1-task (RP-only), 2-task (SCX-RP/hSAX-RP) or 3-task (SCX-hSAX-RP) models but reduced runtime.

8. Utility and Extensibility. The authors spend extensive time determining the utility for a single workflow to investigate an *E. coli* PPI network. How extensible to they believe this platform will be for the general XL-MS user? Can the authors describe how this software could be used with different workflows beyond xiSearch/xiRescore?

Please note that we have now added the analysis of a purified multiprotein complex and herein included the example of using RP data only, as mentioned in our response above (comment 6). RP RTs could be learned for such a small dataset thanks to transfer learning and led to substantial improvements in identified links (1.7-fold for heteromeric links) at constant FDR. This shows that xiRT is not limited to large and multidimensional datasets. Very pleasingly, xiRT is highly useful also to enhance the analysis of multiprotein complexes [3], which is of relevance to the large number of laboratories that use crosslinking for such endeavor.

We appreciate that there are many search workflows beyond that of our lab. Fortunately, and intentionally, xiRT does **not use any search tool specific information**. Due to the very way of how we set up xiRT, it can be taken as a component by the respective developers to complement their existing workflow with retention time prediction. xiRT is an open-source code, stand-alone application that uses tabular search result data. To prove the point, we used the results of pLink2 for our *E. coli* dataset at pLink 1% CSM-FDR as training input for xiRT. As expected, xiRT could learn retention times also from pLink2 (e.g. for RP mean r^2 of 0.9 ± 0 (std)) on the prediction folds not used for training, which demonstrates the general usability of xiRT beyond xiSEARCH (new Fig. S15).

We would have liked to also demonstrate the added benefit of RT information within the pLink2 framework. However, we encountered problems that we duly raised to the developers of pLink2 by opening a Github issue (<https://github.com/pFindStudio/pLink2/issues/80>) on 20 Oct 2020. Unfortunately, we did not receive a response nor was pLink2 updated to this date (5 Mar 2021). This and our experience with Kojak/PeptideProphet (see response to comment 4 above) exemplifies the problems of trying to enhance the workflow of others. Hopefully, our success with retention time prediction and the open availability of xiRT will motivate the developers of other workflows to amend them with retention time prediction soon.

Looking at the larger picture, xiRT was developed for the retention time prediction of crosslinked peptides. We see the strength of xiRT in the following points:

- 1) xiRT works search engine independent. The only required input are CSMs with FDR estimates on CSM level. We now added an evaluation using pLink2 (Fig. S15) to demonstrate this.
- 2) xiRT works with an arbitrary number of RT dimensions. Importantly, even with a single chromatographic dimension (e.g. reversed-phase) and for a purified protein complex, xiRT achieves accurate RT prediction through transfer learning, as we now show (Fig. S10), and thus improves the original search outcome (Fig. 5).
- 3) RT information might benefit the crosslinking field in multiple areas: rescoring, targeted acquisitions, DIA acquisitions, spectral library generation.

We added a note to the conclusion section to communicate the utility more clearly.

9. Expanding Usage. How do the authors envision extending this to human PPI network investigations, e.g. for building an entrapment database or estimating RTs? Including information on this in the Discussion/Conclusions would aid readers' understanding of the utility of this technique outside of the demonstrated use-case.

In this manuscript we focused on the development of a machine learning framework suited for RT prediction of crosslinked peptides. xiRT is agnostic to the workflow / sample / organism and simply requires a collection of CSMs with error estimates in a CSV format. Therefore, xiRT can also be used to predict the retention times of human crosslinks.

There may be a confusion resulting from our use of an entrapment database. The entrapment database was solely used as control **after** the rescoring. As decoys were used during training, we needed a different model for false matches to check the validity of our scoring approach. A separation of decoys and other known false matches (here human target matches) indicates overfitting. Indeed, we observed no increase in known false targets (human targets) after rescoring (Fig. 4b). We would like to emphasize that the entrapment database was not used for the prediction of retention times and is therefore not needed to use xiRT.

10. Heteromeric linkages.

1) Are the authors concerned that the main benefit between xiScore and xiRescore in Figure 4b for CSMs and PPIs is for heteromeric linkages (the claimed 2.1-fold improvement)? As the author's know (Lenz et al. preprint), these linkages are more likely to contain decoy matches compared to self-linkages.

The manuscript by Lenz *et al.* shows that with separate FDR estimation for heteromeric and self-links, the chance for a CSM being wrong is a function of the FDR (and not whether a link is self or heteromeric). Therefore, with proper FDR control, heteromeric links are not more likely to be false / contain decoy matches.

The larger increase in heteromeric links with use of RTs as additional information source indicates that a larger fraction of heteromeric links remain currently unidentified (at a given FDR based on score cut-offs coming from spectrum matching only). In contrast, even with much worse spectra (and correspondingly scores) do self-links pass the same FDR threshold (since there are far fewer possible pairwise peptide combinations for self-links than for heteromeric links; for reference, please see supplementary Fig. 1a in [2]). In essence, after the extensive peptide fractionation employed (nearly) all the self-links in the MS data were already identified while many heteromeric links with MS data were not.

2) The Lenz et al. data also appears to be filtered to a 5% FDR rather than the 1% FDR for xiRescore, though it is used for validation.

In this study, we focused on more stringent FDR estimates because the initial training of xiRT is best done with high-confident identifications with few false positives. In addition, we were interested in how far more CSMs can be identified at lower FDR thresholds. Generally speaking, FDR thresholds are often tightly bound to personal preferences and applications. For example,

Lenz *et al.* uses 5% FDR for showing the principles of FDRs but restricts the FDR to 1% for investigation of PPI interactions later. Both FDR values of 1% and 5% are commonly used in the field (see [2] Table S1).

3) Furthermore, as there are fewer heteromeric human PPIs than CSMS, did the authors observe multiple CSMS within the same human protein pairs? What do they make of these consistent, wrong “interactions”?

We observed only **3 human target CSMS**, resulting in a **single PPI** between a human protein and the *E. coli* protein SucB at 1% PPI-FDR (up to 5% for lower FDR levels). There were no human-human PPIs detected at this FDR. Nevertheless, it is indeed surprising that all 3 human target CSMS matched to a single PPI. When we investigated this in detail, we noticed that all 3 CSMS included the same human peptide (Table R1). This might be a consequence of the (true) second peptide not being in the database (e.g. natural genetic variant). Indeed, closer inspection revealed as a better match the SucB peptide KIKELVAK, i.e. a peptide of the same *E. coli* protein that the link partner is from. It had not been matched as it carries a rare modification that we had not included in our original search. This alternative peptide transforms the heteromeric PPI into a self link, which is probably the correct interpretation.

Table R1: CSMS / PPIs involving a human protein (rescored).

PSMID matches the ID of the results reported by xiSEARCH / xiFDR.

PSMID	Protein1	Protein2 initial	Protein2 corrected	Peptide1 (E. coli)	Peptide2 (human) Initial match	Peptide2 (E. coli) Corrected match
2262348	P0AFG6	P50552	P0AFG6	SEEKcIASTPAQR	KELQKcIVK	KIKcIELVAK
3165715	P0AFG6	P50552	P0AFG6	EDVEKcIHLAK	KELQKcIVK	KIKcIELVAK
2545576	P0AFG6	P50552	P0AFG6	LLAEHNLDAIAIKcIGT GVGGR	KELQKcIVK	KIKcIELVAK

11. RT Comparisons. Other groups have used RT for crosslinked peptide matching (e.g. most recently Steigenberger *et al* MCP [2020]). The authors should provide comparison of whether the 2-fold improvement can be gained through simpler methods – which would be more widely adaptable – to utilize retention time in their analyses.

We would like to emphasize that to the best of our knowledge RT prediction of crosslinked peptides has not been done, prior to our work. Steigenberger *et al.* describes an approach to distinguish mono-linked and cross-linked peptides by measuring collision cross sections in ion mobility-mass spectrometry and then training an SVM classifier “to maximize the separation between mono-linked and cross-linked peptides”. Indeed, we also use an SVM classifier, in our case for rescoring. SVM is one of the simplest approaches one can take and widely used (for example by Steigenberger *et al.*). A filtering-approach might be perceived as even simpler. However, for one, one would have to filter in a high-dimensional space as we have multiple predicted retention time features (also covering the individual peptides). Secondly, we want to integrate the retention dimensions with the mass spectrometric score. SVMs are an established

approach to do this (e.g. Percolator / ProteinProspector / pLink2, albeit none using RT for crosslinks yet).

12. Structure-based FDR. Owing to the large collection of E. coli protein structures, did the authors incorporate any kind of structure-based FDR for estimating intraprotein interactions (Mintseris & Gygi, 2019, PNAS)? There may even be a chance for validation of the heteromeric linkages in well studied complexes.

In this study we focus on the protein-protein-interactions between heteromeric links since these are usually the target in proteome-wide CLMS analysis. We focus the validation on known PPI interactions in (curated) databases such as STRING / APID. Albeit evaluating intra-protein interactions via mapping crosslinks onto structures has been suggested to lead to misleading conclusions [5], we followed the advice of the reviewer to validate the achieved improvements in our multiprotein complex analysis (new Fig. 5), among other quality controls.

13. P8, In 235-238. Can the authors explain why the 4-fold difference was not explained, while the difference observed for the random draw was 3.17-fold?

Please note that “4-fold versus 3.17-fold” is comparing different FDR ranges (1 - 50% and 10 - 50% CSM-FDR). Correct would have been 4- versus 9-fold for our initial data for the full CSM-FDR range. Additionally, please note that we now adapted the sampling process for the random matches, following the comment of another reviewer, and changed the discussion in the text (p.11 “Semi-randomly (...”). However, the fold difference remains the same (4- versus 9-fold). We assume the reviewer wonders about this difference. Therefore, we would like to point out that:

- 1) The now used method is rather pessimistic to estimate the number of expected random matches (one protein from APID/STRING; the other from the fasta database) which might lead to inflated numbers of random matches. However, we think this does not account for the large difference between random and true PPIs. Rather we think the following does:
- 2) There are many more random PPIs possible than true PPIs that are covered by our data. As the score threshold is lowered more and more, random PPIs will pass while the limited number of true PPIs will be exhausted at some point. This is amplified by the tendency of true CSMs to fall into the same PPIs, while random matches will accumulate [2].
- 3) Be this as it may, to us the most important point about this graph (Fig. 1b) is that the absolute number of PPIs that are **correct** and **hidden** in the 50% CSM-FDR data is still much larger than expected by random matching. Here, the absolute differences give a reasonable estimate on the number of missed PPIs at e.g. 50% CSM-FDR. Therefore, it is worth looking for additional information sources (such as RT) that help distinguish true from random matches.

14. Fold improvement. The Conclusion section highlights at 2.1% improvement in total heteromeric crosslinks, but the overall improvement is much more modest. As noted in point 10, these are also the most likely to result from false positives.

As noted above, with separately computed FDR estimates heteromeric links are as trustworthy as self-links, see [1,2]. We would expect the gains to be larger for PPIs, given that the majority of self-links are already identifiable based on MS data alone due to the much smaller search space (see SI Fig.1a in [2]). Note, it is PPIs that we and many others are very interested in.

Minor Concerns

I. Figure 1a is misleading in the depiction of the database usage. The implication is that each spectra is searched against full databases and these are processed by BLAST/Comet to generate the final dataset.

We adapted the figure to better represent the actual workflow and corrected the tool names.

II. P1, In 19. The margin of error comment is a little misleading as it really represents 11% windows around the fractions. So, with 20 fractions, this would be +/- 2 fractions, etc.

We used the relaxed accuracy (racc) metric of +/-1 error on each prediction task independently. For each task (SCX, hSAX), this metric is estimated on the prediction folds from multiple CSMs. According to the used definition, a racc of 90% follows the intuition that in 90% of the cases the margin of error is within +/-1 between the observed and predicted fraction.

III. P1, In 22. The claim "...2-fold increase in protein-protein interactions detected..." is misleading based on the actual results presented in Figure 4b.

We have adapted the text and references to that figure accordingly.

IV. P2, In 38. There have been several enrichable linkers developed prior to the PhoX linker and should be referenced. The authors should also mention chromatographic enrichment (e.g. SCX), and IMS enrichment strategies (Schnirch et al Anal Chem [2020], Steigenberger et al, MCP [2020]). All these methods are potentially complementary.

We have improved the coverage of and indeed focused on chromatographic methods in crosslinking MS in the introduction while covering other methods now through citing reviews.

V. P2, In 46. The claim of "...multi-dimensional chromatography workflows can yield in the order of 10,000 CSMs at 1-5% false discovery rate (FDR)..." should have a reference associated with it or be removed.

We added four references in which more than 10,000 CSMs were identified.

VI. The author's consistently use lower case lettering for acronyms (e.g. "blast" in Figure 1a) and the first letter of named programs (e.g. "comet" [p5, In104]). These should be changed throughout the figures and manuscript to the correct usages.

We corrected the spelling of the mentioned programs and acronyms.

VII. The authors set up “three major challenges” in the opening paragraph (p2, ln 28-36), but do not mention how their new pipeline solves the issues of low abundance, unequal fragmentation, or combinatorial complexity. Could they provide details on where they think the current work fits into solving these central issues?

We restructured parts of the abstract and introduction to make it clearer how RT information can help with the mentioned challenges and come back to this in the conclusion.

VIII. The “Sample Preparation” section of the Methods makes no mention of reducing the proteins prior to alkylaton. Was this step performed?

The sample preparation in this manuscript was kept rather short, however, it follows the exact same protocol as described in Lenz *et. al* [2]. We added the missing reduction step to the sample preparation section.

Reviewer #3 (Remarks to the Author):

The manuscript by Giese et al. describes a machine learning algorithm for the prediction of crosslinked peptide retention times under multidimensional fractionation techniques. When using the difference between these predictions and the observed retention times additional parameters were produced that improved the number protein-protein interactions observed at various FDR thresholds. These additional interactions shown to be plausible when compared to entrapment interactions, suggest that RT prediction is a valuable tool to improve crosslink search results over search score alone.

The concepts in the paper are interesting, and parallel beneficial efforts that were made in single peptide identification and validation. The authors provide their software tool (xiRT), which appears to be generalizable to workflows outside their lab, and thus a widely useful tool to the community. Considering that crosslinked peptide identification has typically been fraught with false identifications, and that current efforts have largely focused on controlling the error by adopting stricter thresholds, this work perhaps promises to improve analyses through retention of more [correct] identifications following validation. I see this research as having a positive impact on the crosslinking community.

The manuscript does contain several points that need clarification prior to publication:

1. xiRT RT prediction appears to be based on DePART, which is merely referenced in the manuscript. I feel this is a critical component that requires at least a brief overview, even if in the supplementary information.

We added a note to the “xiRT - 3D Retention Time Prediction” methods section (p.7).

2. 3D fractionation does not appear to be commonplace to me. Most crosslinking research seems to use simply RP fractionation, and possibly the addition of SCX. Though the online instructions for xiRT possibly indicate it works with RP-only data, the manuscript is not clear on this fact. How much benefit could be expected if simpler fractionation schemes are used (SCX-RP or RP only, for example)? This might be highly relevant considering the RP model struggled with TD matches (see line 322).

Many large-scale studies have shifted to use at least one additional retention dimension. xiRT works with an arbitrary number of chromatographic retention dimensions. We have added analyses using RP only, SCX-RP and hSAX-RP in the *E. coli* data to the supplementary Material (Fig. S6). In any case, we now performed the entire analysis also on a multiprotein complex, the FA-complex [3] where we only used the RP dimension. In this set-up, the RP information alone helped to increase the number of detected crosslinks (new Fig. 5).

3. I found the SHAP analysis difficult to follow.

- (a) feature interactions were designated as ';' in the supp_info, but with '*' in the main text (and described with ';' in the caption).

- (b) is the use of '*' to indicate that these interactions are actually each predictor multiplied together and treated as a parameter?

We completely reworked the feature generation and simplified the supplementary material resolving the mentioned concerns. The complete list of features is now given in Table S3.

(c) how was only 10 features (of 130) chosen to illustrate importance of various features? I find it unlikely to assume that all 130 are similar.

Showing 10 features was a choice made to simplify the interpretation of the results. These 10 features were detected as most important by the magnitude of the SHAP value. As shown in Figure 3c, the magnitude of the SHAP value is already quite small for the 10th feature. Therefore, the following features will have an even smaller impact on the model prediction. Note that we revised the feature set in the revised manuscript and ended up with a smaller number of features overall. In any case, we increased the number of shown features to 15.

(d) how come there appears to be a large difference in the top 10 features for each SHAP plot provided (1 in main text, 2 in supp_info)? Might expanding the analysis beyond 10 features help identify features common among each analysis, and presumably most important to the model?

Based on the reviewers concerns we have simplified the presentation of the SHAP results. We now limit the presentation to the TTs that were not used during training with an SVM-score > 0.5 to focus on the features that help detecting previously unidentified TTs. In response to the question, given the old results: the beauty of SHAP lies within the ability to locally explain feature importance, i.e. analyze the feature importance of individual observations like a specific CSM. For the presented SHAP analysis we previously used very different subsets a) TTs with an FDR $> 1\%$ (main text), b) 50 TT / 50 TD that with an FDR $\leq 1\%$ and c) TDs with an FDR $> 1\%$. Therefore, SHAP revealing different sets for the top ten features of these three data subsets met our expectations.

4. Machine learning algorithms benefit most from large datasets. The datasets here are large, far larger than many publications (144 acquisitions vs. a dozen or fewer in many publications I've seen). The authors noted the limitation and performed analyses on subsets to model the effect. But I feel this effort stopped short of providing realistic context to the readers.

- 1) What would someone do if they had only 100 CSMs?
- 2) Can the authors actually recommend minimum dataset sizes for reliable xiRT performance?
- 3) And if so, are they still applicable when using fewer features (such as performing the analysis on RP-only data)?

Already a few years ago, one could obtain ~1,000 CSMs from three replicate acquisitions of a single crosslinked protein (e.g. [6]). Nowadays, one can certainly achieve more in a typical analysis of a multiprotein complex. Crosslinking MS data acquisition of challenging samples generates a lot of data (see Table S1 in [2]). Here our method can be used best. However, smaller samples can still be used through the application of transfer-learning.

We tested xiRT with a much smaller data set from Shakeel *et al.* [3], a purified multiprotein complex with 20 acquisitions (BS3-crosslinking data set). This is an example for a very routine experiment in structural biology of protein complexes done nowadays. For the FA-complex, we identified 1376 CSMs (TT: 1059 CSMs, 307 heteromeric CSMs) at 5% CSM-/peptide-pair FDR and 1% residue-pair FDR. When we trained xiRT on CSMs filtered to 1% CSM-FDR, the model could not converge and only resulted in an r^2 of 0.49. We would tend to call this a failure.

However, when we pretrained xiRT on the data in this manuscript (DSS-crosslinked) we achieved an average r^2 of 0.91 with weight-adjustment (new Fig. 5 and S10). So, xiRT also works for much smaller datasets and with a single chromatographic dimension (RP).

5. I attempted to use xiRT and hit two roadblocks that can probably be fixed by improving the online tutorial. (a) xiRT aborted because `libcuda.so.1` could not be found. This is because I don't have an NVIDIA card, and I thought from the documentation that CUDA was optional. If not, this needs to be explicitly stated. (b) xiRT config and setup config are not documented. While I think I could navigate one file correctly, I cannot make sense of the parameters listed in the other. Both of these files need clear documentation and tutorials, not simply a link to YAML.

We have greatly improved the documentation on Github¹, as well as the general documentation². The improved documentation shows a slightly improved installation guide that should fix the mentioned cuda problem (we recommend using conda for the TensorFlow installation). The reviewer is correct that xiRT can be used either with CPU or GPU. However, the respective option must be given in the parameter file for which we improved the documentation as well³. In addition, we added a couple of example parameterization of xiRT to better reflect the possible use cases⁴.

¹ <https://github.com/Rappsilber-Laboratory/xiRT>

² <https://xirt.readthedocs.io/en/latest/>

³ <https://xirt.readthedocs.io/en/latest/parameters.html>

⁴ <https://xirt.readthedocs.io/en/latest/usage.html#examples>

Reviewer #4 (Remarks to the Author):

xiRT did a good job in retention time (RT) prediction for cross-linked peptides (CX). This is certainly a novel method since there are no RT prediction tools for CXs as I know so far. There are 3 main contributions for xiRT as shown in its abstract: (1) Model CXs by using Siamese Net and using multi-task learning for SCX, hSAX, and RT prediction; (2) It is quite accurate for SCX, hSAX, and RT prediction; (3) Percolator-like rescoring based on predicted SCX, hSAX, and RT features would significantly increase the PPI detections at a proteome-scale (E. coli lysate). This work will be useful for CX-MS analysis.

Here are my comments:

1. It is a great idea to use Siamese Net to model CX problems. Although I do not work on the pLink project, I have a similar idea for predicting "something" of CXs, but this work moved faster than us.
2. Multi-task learning (MTL) is also a good idea. My question is, although MTL can save the training and predicting time,
 - 1) I wondered if single-task learning can achieve a better performance?
 - 2) And what the common knowledge did the model learn within the shared layers? The latter question may be too difficult to answer, therefore, at least, authors should show MTL is necessary for higher prediction accuracies.

1) The Reviewer raises a valid question which we now answer in the supplementary material (Figure S6 and S7). In short, the performance is very comparable for single-task (ST) and multi-task (MT) parameterizations and does not yield significant performance differences (ANOVA). However, the overall runtime of xiRT is greatly decreased (~1/3 on both CPUs and GPUs). This is extremely valuable for hyper-parameter optimization of the neural network. Note that we limited our analysis to 50% CSM-FDR which also greatly decreases runtime. Future developments, especially when rescoring workflows are applied after retention time prediction might need to use all identified CSMs for model building.

2) In addition, we have investigated the learned features from the network with two approaches:

a) We investigated the intermediate layers of the neural network by applying dimensionality reduction for each of the layers using UMAP. Interestingly, the RNN-layer mostly catches the RP features, while SCX / hSAX features are only weakly separated (Fig. S14).

b) We have extended the usage of the SHAP package to the peptide sequences. However, SHAP has some current limitations with TensorFlow > 2.0, which does not support multi-task learning (natively) as well as ordinal feature encoding. This leads to larger errors in the estimated importance values via SHAP. Given their approximate nature, we limit ourselves to a description of the SHAP values in the supplementary information. E.g. feature attributions can be retrieved per residue in a crosslinked peptide (Fig. S11, Fig. S12) revealing insights into the (crosslinked) residue contributions towards retention time. This can be extended to global analysis (Fig. S13) that can improve the general understanding of retention mechanisms for crosslinked peptides. Importantly, this analysis is not necessarily limited to retention times. The recent interest in ion-mobility separation for crosslinked peptides is another interesting use case

for xiRT and the applied explanation model. To the best of our knowledge this is also the first approach to use explainable AI techniques for retention time prediction and crosslinked peptides.

3. After the model is designed, it is not hard to build deep neural network models for RT prediction using traditional regression techniques, but it is interesting that xiRT uses ordinal regression models for SCX and hSAX prediction.

- 1) But there is a concern here: the number of fractions should be fixed for deep ordinal regression network, making it difficult to extend for different fraction numbers?
- 2) How do the authors consider this problem?

In our opinion, two possible solutions exist:

- a) Neural networks offer the possibility to utilize already trained architectures and repurpose the trained model. As can be seen in the response to question 2 of this reviewer, the neural network already has very valuable information about the input data in the first layers. Therefore, in this scenario, the most appealing approach would be to remove the last layer from the initial trained model that holds the information how to combine the features into the fraction predictions. In this approach, arbitrary changes in the fractionation design are possible. Based on the reviewer's concern, we added an option to xiRT to make the use of this scenario easier for the user (xiRT v. >1.1.0). This approach will require moderate retraining of the final layers of the network.
- b) xiRT also offers the possibility to use standard regression methods for the prediction of fractionation data by changing the activation function in the last (prediction) layer. While theoretically not optimal, this solution is easier to use and transform.

4. At line 109, I suggest to put the sentence "Before the identification with xiSEARCH the masses of precursor and fragment ions were recalibrated." into the front of the xiSEARCH settings, otherwise it will be confused that why fragment tolerance is only 5 ppm.

We thank the reviewer for the comment and adapted the text accordingly.

5. At line 144, the authors said that "The input of xiRT are amino acid sequences with arbitrary modifications", but at the next line, it said "... encoded by replacing every amino acid by an integer", obviously the encoding did not really take "arbitrary" modifications into account, at least I did not find it in this paper.

We clarified the respective text in the manuscript. "Arbitrary" here means that no peptide modifications from text format are excluded. The integer encoding is done on-the-fly for the entire alphabet in the input data (see paragraph "xiRT - 3D Retention Time Prediction").

6. At line 186, xiRT uses rbf kernel for its Percolator-like algorithm.

1) Is it necessary to use a non-linear kernel instead of a linear one? As we know, Percolator uses the linear kernel. Besides, the features (score, RT difference, etc.) used by xiRT may have linear properties. The on-linear kernel tends more easily to be overfitting, especially for some scenarios such as CX search which has fewer training samples.

We thank the reviewer for the interesting question. While we did not observe overfitting in the first submission of the manuscript (judged by increase of entrapment hits after the rescoring), we decided to use linear SVM since it is much faster and, as the reviewer points out, more robust against overfitting. Together with a drastically changed feature set, we still achieve improvements on CSM- and PPI-level on complex *E. coli* data (Fig. 4), as well as a much smaller data set from a single complex (FA-complex, Fig. 5). In both cases we use the hits to the entrapment database before / after the rescoring as control. Since the entrapment TT hits are treated as TTs they should not increase after the rescoring and in fact they did not.

7. At line 190, I don't understand why there are so many (130) features.

We adapted the methods part explaining the feature generation. The number of features results from the crosslinked peptide predictions for the three RT dimensions (3 features), individual peptide predictions (3 features for peptide1 and 3 features for peptide2) and multiple other features based on this information. We have redefined the initial feature generation; please refer to Table S3 to get the full list of used features.

8. From line 191 to 193, does it mean all DXs are negative samples? If so, this description may be too complicated to understand. Otherwise, what did the last sentence suppose to mean?

We revised the text to better explain that

- 1) TTs were treated as positively labeled observations and target-decoy and decoy-decoy identifications were labeled as negative observations.
- 2) only TTs $\leq 1\%$ CSM-FDR were used, while all TD/DD regardless of the FDR were used.

9. At line 200, xiRT uses a 3-fold CV for **rescoring steps**. It is reliable, but my question is about extensibility. Most of the structural biologists may only concern about protein-complex level identification instead of proteome-wide level identification, which means there may be not that enough CX-peptides for 3-fold CV for protein-complex identification

We now show on data of a purified multiprotein complex that xiRT can predict retention times (thanks to transfer learning) and that these retention times positively affect the identification success (see response to Reviewer 3, comment 2). We added this analysis as new Fig. 5 and Fig. S10.

10. Line 233, it is a good idea to use PPI-level evidence to validate CX identifications, but the problems are how to estimate the PPI-level error rate itself? How PPI-error-rate increases as the CSM-FDR increases?

For the error estimation on PPI-level we used the tool xiFDR [1]. xiFDR was specifically developed for the error estimation for Crosslink MS data and deals with the several "levels" of biological information with individual FDR estimates (spectrum, peptide-pair, residue-pair, protein-pair) - very similar to the peptide and protein FDR in proteomics. As the reviewer points out, this level-specific FDR estimation is important since an increase in CSM-FDR can lead to

significant increases on the PPI-FDR [1]. Our lab is introducing a very rigorous assessment of PPI-FDR estimation in a recent manuscript [2] which was implemented already in xiFDR and consequently also used here.

11. Line 236, "Randomly drawing pairs of E. coli proteins" may be a wrong simulation to evaluate the false negatives at 10% or 50% CSM-FDR. As we all know, there are a lot of True-False-linked CXs (i.e. half-correct CXs) at x% CSM-FDR, hence the pairs are not totally random, maybe they are mostly "half-random". I suggest authors should re-consider this simulation.

The reviewer raises an interesting point that we have not considered in our previous approach. We followed the reviewer's suggestion and implemented the mentioned strategy, i.e. the first protein is randomly drawn from the unique set of proteins in STRING/APID. The second protein is drawn from the FASTA file. This approach resulted in an increase in random matches in the respective databases, e.g. for 50% CSM-FDR the number of random matches increased from initially 54 random matches to 91 with the revised semi-random sampling strategy. Importantly, the number of "validated" PPIs still leaves room for many true PPIs to be discovered and secondly, the now used sampling strategy is rather pessimistic while the previous one was optimistic (conservative). The truth lies probably between the two estimates.

12. Line 152, it said that xiRT uses an additive layer, but it said: "Multiply-layer was used" in "Siamese Architecture" in Supporting Information S1, it might be a small mistake.

We adapted the text and Fig. S1 to match the best hyperparameters from the manuscript. Previously, the figure was showing a possible parameterization of xiRT since the "combination"-layer after the Siamese networks can be chosen from a set of predefined options (add, concat, multiply, maximum, average).

13. Fig. 1d, This network illustration might be not 'siamese' enough, readers cannot get what is the siamese net from this figure if they have not heard about it.

We thank the reviewer for the comment and reworked the figure and the caption to help readers to understand the Siamese-part more intuitively.

14. Fig. 4b, I have two questions here:

a. What is the PPI FDR? I've never heard about it.

PPI-FDR stands for protein-protein interaction FDR. As detecting PPIs is the target of many crosslinking studies (certainly if done at large scale), we use the information on how many PPIs pass a pre-defined FDR threshold as a key evaluation metric. For the error estimation on PPI-level we used the tool xiFDR [1]. xiFDR was specifically developed for the error estimation in crosslinking MS data and deals with the several "levels" of biological information (spectrum, peptide-pair, residue-pair, protein-pair) with customized FDR estimates. See response to comment 10, above.

b. For the increment of the xiRescore, the proportion of PPI identifications are significantly larger than that of CSMs, does these additionally identified PPIs are “one-hit-wonder”? Or how to evaluate the quality of additionally identified PPIs?

Following a reviewer's comment, we now chose a more conservative approach for rescoring. The proportional gains in CSMs are now larger than those in PPIs (compare Fig. 4b). Most new CSMs resulting from rescoring fall into PPIs that were observed with multiple CSMs (Fig. 4d). Note also, that not only the number of CSMs changes but also their score which has a direct impact on the FDR estimation. The merging of CSM-level information into peptides pairs, residue pairs or protein interactions comes with a change in score in the used software xiFDR. E.g. multiple CSMs can represent the same crosslinked peptide pair and the score for that peptide pair is derived by: $S_{PP} = \sqrt{\Sigma(S_{PSM} \times S_{PSM})}$, where S represents the score for the respective level [2, SI]. Therefore, a change of score in the lowest level (CSM) can have a large impact on the other scores (peptide-pair, residue-pair and PPI).

In addition, we perform three validation checks for the newly identified PPIs:

- 1) Comparison to a much larger study (*Lenz et. al.*)
- 2) Comparison to public databases (STRING / APID)
- 3) Indirect quality assessment by comparing to the number of human PPIs. Since the number of these human PPIs is very low, the statistical power is low. However, a low number of human PPIs is of course desirable and could indeed be achieved after the machine learning step.

References:

- [1] Fischer, L., & Rappsilber, J. (2017). Quirks of Error Estimation in Cross-Linking/Mass Spectrometry. *Analytical Chemistry*, 89(7), 3829–3833. <https://doi.org/10.1021/acs.analchem.6b03745>
- [2] Lenz, S., Sinn, L. R., O'Reilly, F. J., Fischer, L., Wegner, F., & Rappsilber, J. (2020). Reliable identification of protein-protein interactions by crosslinking mass spectrometry. *BioRxiv*, 2020.05.25.114256. <https://doi.org/10.1101/2020.05.25.114256>
- [3] Shakeel, S., Rajendra, E., Alcón, P., O'Reilly, F., Chorev, D. S., Maslen, S., Degliesposti, G., Russo, C. J., He, S., Hill, C. H., Skehel, J. M., Scheres, S. H. W., Patel, K. J., Rappsilber, J., Robinson, C. V., & Passmore, L. A. (2019). Structure of the Fanconi anaemia monoubiquitin ligase complex. *Nature*, 575(7781), 234–237. <https://doi.org/10.1038/s41586-019-1703-4>
- [4] Beveridge, R., Stadlmann, J., Penninger, J.M. *et al.* A synthetic peptide library for benchmarking crosslinking-mass spectrometry search engines for proteins and protein complexes. *Nat Commun* 11, 742 (2020). <https://doi.org/10.1038/s41467-020-14608-2>
- [5] Yugandhar, K., Wang, TY., Wierbowski, S.D. *et al.* Structure-based validation can drastically underestimate error rate in proteome-wide cross-linking mass spectrometry studies. *Nat Methods* 17, 985–988 (2020). <https://doi.org/10.1038/s41592-020-0959-9>
- [6] Giese, S. H., Belsom, A., & Rappsilber, J. (2016). Optimized fragmentation regime for diazirine photo-cross-linked peptides. *Analytical Chemistry*, 88(16), 8239–8247. <https://doi.org/10.1021/acs.analchem.6b02082>
- [7] Steigenberger, B., Albanese, P., Heck, A. J. R., & Scheltema, R. A. (2020). To cleave or not to cleave in XL-MS? *Journal of the American Society for Mass Spectrometry*, 31(2), 196–206. <https://doi.org/10.1021/jasms.9b00085>

REVIEWERS' COMMENTS

Reviewer #2 (Remarks to the Author):

NCOMMS-20-28567A – Revision

The author's present a much-revised manuscript and figures that improve the explanation and benefits of their methods. The novelty of the RT prediction and the improvements, particularly in heteromeric crosslink identifications remains impressive. The addition of 1-, 2-, and 3-task modeling and the new comparisons using RP-only and purified complex analyses add nice evidence for the utility of xiRT. Additionally, the new SHAP plots and surrounding work are quite nice.

Upon reading the revised manuscript, I have some minor comments, but I believe this work is in good shape for publication.

Comments

1. The authors mention using their DSS xiRT predictions for a BS3 dataset (Shakeel et al.). The resulting links are chemically identical. Do the authors believe it would be possible to use the training data in this work to predict RTs for more complex crosslinkers? For example, how much is the hydrocarbon linker affecting the prediction results versus the physicochemical properties of the amino acids themselves?
2. To clarify the intended question regarding heteromeric linkages, rather than "...these linkages are more likely to contain decoy matches...", the question should have referenced the point in the Lenz et al. manuscript that "...most false positives in the total (self and heteromeric) set of crosslinks will be heteromeric." The response that "...after the extensive peptide fractionation employed (nearly) all the self-links in the MS data were already identified while many heteromeric links with MS data were not" seems somewhat controversial.
 - a. Are the authors arguing that the lower score thresholds needed for 'accurate' identification of self-links? If that were true how do the authors explain the relatively few total proteins with self-links compared to all potential NHS-reactive residues in close proximity within a given cell or lysate? Or do the authors mean "identifiable crosslinks"?

b. Alternatively, are the authors on the verge of speculating that they can do even better? E.g., using gas phase fractionation or IMS?

3. The results described in Table R1 are quite interesting, and I would encourage the authors to include them and the description of these 3 human target CSMs.

Reviewer #3 (Remarks to the Author):

All my concerns have been addressed and I consider the manuscript ready for publication.

Reviewer #4 (Remarks to the Author):

Almost all of my concerns have been addressed, and obviously, the authors did more work than I expected, thank you very much. I think the manuscript can be published as it is. Congratulations to the authors for the new useful tool in CXMS.

For further discussion, I am still not sure about the reliability of false-negative estimation in Fig. 1b. I am not sure if all PPIs in STRING/APID database should always be presented in different samples regardless of conditions (different growing states, different environments, or even different cells). Or are PPIs dynamically changing? If all PPIs are always presented regardless of conditions, then we can use them as positive controls. Otherwise, if only 50% PPIs are presented, we will lose 50% reliability. For example, we cannot say proteins are reliably identified because they are from the reviewed UniProt database. I am not familiar with PPIs as I am from computer science.

This concern is not to say that improved identifications from xiRT are unreliable. But I think false-negative estimation is still an open problem.

Dr. Wen-Feng Zeng

REVIEWER COMMENTS

Reviewer #2 (Remarks to the Author):

NCOMMS-20-28567A – Revision

The author's present a much-revised manuscript and figures that improve the explanation and benefits of their methods. The novelty of the RT prediction and the improvements, particularly in heteromeric crosslink identifications remains impressive. The addition of 1-, 2-, and 3-task modeling and the new comparisons using RP-only and purified complex analyses add nice evidence for the utility of xiRT. Additionally, the new SHAP plots and surrounding work are quite nice.

Upon reading the revised manuscript, I have some minor comments, but I believe this work is in good shape for publication.

Comments

1. The authors mention using their DSS xiRT predictions for a BS3 dataset (Shakeel et al.). The resulting links are chemically identical. Do the authors believe it would be possible to use the training data in this work to predict RTs for more complex crosslinkers? For example, how much is the hydrocarbon linker affecting the prediction results versus the physicochemical properties of the amino acids themselves?

Yes, we believe this is possible. A large part of what the network learned are contributions of the (non-crosslinked) amino acids. Adjustments for crosslinked residues are usually only necessary on a subset of the network's weights (depending on crosslinker specificity). The choice of the linker will likely have an impact on an analyte's retention time, however we cannot quantify this influence yet. Within xiRT, transfer learning between datasets that use crosslinkers with different backbones should be straightforward, given similar chemical reactivity / site specificities.

2. To clarify the intended question regarding heteromeric linkages, rather than "...these linkages are more likely to contain decoy matches...", the question should have referenced the point in the Lenz et al. manuscript that "...most false positives in the total (self and heteromeric) set of crosslinks will be heteromeric." The response that "...after the extensive peptide fractionation employed (nearly) all the self-links in the MS data were already identified while many heteromeric links with MS data were not" seems somewhat controversial.

a. Are the authors arguing that the lower score thresholds needed for 'accurate' identification of self-links? Comparing the score threshold of self-links with that of heteromeric linkages clearly shows that the former is much lower than the latter for the same FDR. To us, this appears to be an observation made on the data rather than a point to argue.

If that were true how do the authors explain the relatively few total proteins with self-links compared to all potential NHS-reactive residues in close proximity within a given cell or lysate?

To us, there appears to be a confusion of points of views. We are looking at those links that can be found in our mass spectrometric data. There are likely many other links in the sample, below the detection limit of our mass spectrometric acquisition.

Or do the authors mean "identifiable crosslinks"?

Yes indeed, if we define “identifiable crosslinks” as those that surpass the detection limit of our mass spectrometric acquisition, i.e. are covered by data.

B. Alternatively, are the authors on the verge of speculating that they can do even better? E.g., using gas phase fractionation or IMS?

When (insufficient) mass spectrometric evidence is limiting the identification of peptides, almost any kind of additional information will help in the identification process.

xiRT can also use features from IMS such as collisional cross section.

3. The results described in Table R1 are quite interesting, and I would encourage the authors to include them and the description of these 3 human target CSMs.

We added the data as a supplementary table 6.

Reviewer #3 (Remarks to the Author):

All my concerns have been addressed and I consider the manuscript ready for publication.

Reviewer #4 (Remarks to the Author):

Almost all of my concerns have been addressed, and obviously, the authors did more work than I expected, thank you very much. I think the manuscript can be published as it is. Congratulations to the authors for the new useful tool in CXMS.

For further discussion, I am still not sure about the reliability of false-negative estimation in Fig. 1b. I am not sure if all PPIs in STRING/APID database should always be presented in different samples regardless of conditions (different growing states, different environments, or even different cells). Or are PPIs dynamically changing? If all PPIs are always presented regardless of conditions, then we can use them as positive controls. Otherwise, if only 50% PPIs are presented, we will lose 50% reliability. For example, we cannot say proteins are reliably identified because they are from the reviewed UniProt database. I am not familiar with PPIs as I am from computer science. This concern is not to say that improved identifications from xiRT are unreliable. But I think false-negative estimation is still an open problem.

Dr. Wen-Feng Zeng

We agree with the reviewer in that a PPI tentatively observed in an analysis is not automatically correct simply because it has been listed as a PPI before, e.g. in STRING/APID. In addition, the detectability of PPIs will change depending on growth states and other environmental stimuli etc.

Importantly, we are not arguing about exact numbers of false negatives and use STRING/APID only as an estimate for false negatives. The substantial number of PPIs from STRING/APID seen among our negatives is not explicable by random drawing. Consequently, we must miss a considerable number of false negatives in our analysis when using MS evidence alone for decision-making. This motivated our use of retention time as an additional information source.